



# Evaluation of Sea-Ice Thickness from four reanalyses in the Antarctic Weddell Sea

Qian Shi[1,2], Qinghua Yang[1,2], Longjiang Mu[3,2], Jinfei Wang[1,2], François Massonnet[4], Matthew Mazloff[5]

[1]School of Atmospheric Sciences and Guangdong Province Key Laboratory for Climate Change and Natural Disaster Studies,
Sun Yat-sen University, Zhuhai 519082, China
[2]Southern Marine Science and Engineering Guangdong Laboratory (Zhuhai), Zhuhai 519082, China
[3]Alfred Wegener Institute Helmholtz Centre for Polar and Marine Research, Bremerhaven 27570, Germany
[4]Georges Lemaître Centre for Earth and Climate Research, Earth and Life Institute, Université catholique de Louvain, Louvain-la-Neuve, Belgium
[5]Scripps Institution of Oceanography, University of California, San Diego, CA, USA

*Correspondence to*:  Yangqh25@mail.sysu.edu.cn and Longjiang.mu@awi.de

**Abstract.** Ocean-sea ice coupled models constrained by varied observations provide different ice thickness estimates in the Antarctic. We evaluate contemporary monthly ice thickness from four reanalyses in the Weddell Sea, the German contribution of the Estimating the Circulation and Climate of the Ocean project, Version 2 (GECCO2), the Southern Ocean State Estimate
(SOSE), the Nucleus for European Modelling of the Ocean (NEMO) based ocean-ice model (called NEMO-EnKF), and the Global Ice-Ocean Modeling and Assimilation System (GIOMAS), and with reference observations from ICESat-1, Envisat, upward looking sonars and visual ship-based sea-ice observations. Compared with ICESat-1 altimetry and in situ observations, all reanalyses underestimate ice thickness near the coast of the western Weddell Sea, even though ICESat-1 and visual observations may be biased low. GECCO2 and NEMO-EnKF can well reproduce the seasonal variation of first-year ice
thickness in the eastern Weddell Sea. In contrast, GIOMAS ice thickness performs best in the central Weddell Sea, while SOSE ice thickness agrees most with the observations in the southern coast of the Weddell Sea. In addition, only NEMO-EnKF can reproduce the seasonal spatial evolution of ice thickness distribution well, characterized by the thick ice shifting from the southwestern and western Weddell Sea in summer to the western and northwestern Weddell Sea in spring. We infer that the thick ice distribution is correlated with its better simulation of northward ice motion in the western Weddell Sea. These
results demonstrate the possibilities and limitations of using current sea-ice reanalysis for understanding the recent variability of sea-ice volume in the Antarctic.

## 1 Introduction

Antarctic sea ice is a crucial component of the Earth system. In contrast to the rapid sea ice decline in the Arctic, the sea-ice extent of the Antarctic exhibited an overall positive trend during the past four decades (Simmonds, 2015; Comiso et al., 2017),
even when taking into consideration the relatively fast decrease observed from 2014 to 2017 (Turner and Comiso, 2017; Parkinson, 2019). Potential causes such as the ozone hole (Thompson, 2002; Turner et al., 2009), the interactions of the





atmosphere and ocean (Stammerjohn et al., 2008; Meehl et al., 2016), and the basal melting from ice shelves (Bintanja et al., 2013) have been proposed to explain this phenomenon, but a consensus has not yet been reached (Bitz and Polvani, 2012; Sigmond and Fyfe, 2014; Swart and Fyfe, 2013; Holland and Kwok, 2012). Due to limited ice thickness measurements,

previous investigations primarily focused on the change of sea-ice extent or area rather than sea-ice volume. However, sea-ice thickness, which determines the sea ice storage of heat and freshwater, is a significant parameter meriting further investigation. Understanding the causes of changing sea-ice thickness is vital for both understanding the sea-ice mass change over the past decades and predicting the sea ice change in the Antarctic (Jung et al., 2016).

The significant role of the Weddell Sea in sea ice formation (accounting for 5~10% of sea-ice volume, see Tamura et al., 2008) makes the region a significant source of Antarctic Bottom Water (AABW) formation (Gill, 1973). Decreasing sea ice production in the Weddell Sea will therefore further freshen AABW (Jullion et al., 2013). Apart from the seasonal sea ice, the Weddell Sea also has perennial sea ice (about $1\times10^6$ km$^2$, accounting for 40% of the total summer sea-ice area in the Antarctic). This perennial sea ice is found in the northwestern Weddell Sea along the Antarctic Peninsula (AP), due to the semi-enclosed

basin shape and the related clockwise gyre circulation (Zwally et al., 1983). The extent of the perennial sea ice influences radiation and momentum budgets of the upper ocean in the summertime. Moreover, the Weddell Sea is found to be the main contributor to the increasing Antarctic sea ice volume trend in results from different models (Holland et al., 2014; Zhang, 2014).

Unlike in the Arctic, sea ice thickness observations, such as those from submarines or airborne surveys (Kwok and Rothrock, 2009; Haas et al., 2010) are rather sparse and rare in the Antarctic. Drillings offer ice thickness information on level or undeformed ice but are not representative of large scales. Before 2002, large-scale Antarctic sea-ice thickness observations mainly came from visual measurements on ships, such as those provided by the Antarctic Sea Ice Processes & Climate program (ASPeCt) (Worby et al., 2008). The ASPeCt data are valuable for undeformed ice and thin ice, but have obvious negative

biases and do not inform the ice thickness during wintertime (Timmermann, 2004). Ice draft from upward looking sonar (ULS) can be used to investigate ice evolution, but the deployments are mostly in the Weddell Sea. Recently, autonomous underwater vehicles (AUV) carrying ULS devices becomes a novel method to collect the contemporary wide sea-ice draft map. Williams et al. (2015) indicated that the Antarctic inner ice is likely more deformed than previously thought based on ULS observations aboard AUV. However, the application of AUV ULS is still limited to regional observational efforts. Since the launch of a

laser altimeter aboard ICESat-1 and radar altimeters aboard Envisat and CryoSat-2, basin-wide sea ice thickness can be estimated (Zwally et al., 2008; Kurtz and Markus, 2012; Yi et al., 2011; Hendricks et al., 2018). The Antarctic sea-ice thickness from ICESat-1 has already been widely used in the Antarctic sea ice research but it is also reported to have uncertainties due to the poor knowledge of the snow cover (Kurtz and Markus, 2012; Yi et al., 2011). Moreover, the relatively short temporal coverage of ICESat-1 (totally 13 months from spring to autumn) impedes its application for climate studies. Envisat and

CryoSat-2 cover longer time periods (from 2002 to present), but they tend to overestimate Antarctic thickness resulting from



the uncertain representation of snow depth (Willatt et al., 2010; Wang et al., 2020). In addition, current altimeters only provide sea-ice thickness maps over the whole Arctic or Antarctic once a month due to their relatively narrow footprints. It is worth noting that Antarctic IceBridge data can provide ice thickness during summertime based on aerial remote sensing since 2009 (Kwok and Kacimi, 2019).


Compared with sea-ice thickness from in situ or remote sensing observations, thickness estimates from reanalysis systems have advantages with regards to the temporal duration and spatial coverage. However, fully coupled climate models display large systematic biases because of the complex atmosphere-sea ice-ocean feedbacks (e.g., Zunz et al., 2013). In view of these biases, the use of sea ice-ocean models forced by atmospheric reanalysis is a general approach to better constrain sea-ice

thickness changes. In addition, the data assimilation is an effective approach to further narrow the gap between model simulations and observations. Several investigations estimate long-term Antarctic sea-ice thickness using ice-ocean coupled models with data assimilation (e.g., Zhang and Rothrock, 2003; Massonnet et al., 2013; Köhl, 2015; Mazloff et al., 2010). These sea-ice thickness products have been used for various studies. However, to our knowledge, there have been no comprehensive inter-comparisons conducted on these data sets, particularly in the Weddell Sea.


Different from the other Antarctic marginal seas, the Weddell Sea fortunately has more in situ sea-ice thickness measurements, including moored upward looking sonar and drillings (Lange and Eicken, 1991; Harms et al., 2001; Behrendt et al., 2013). In this paper, we evaluate four widely used Antarctic sea-ice thickness reanalysis products in the Weddell Sea against most of the available ice thickness observations. We focus on the inter-comparison of the sea-ice thickness performance and do not

attempt to find the causal mechanisms for the spread in the data sets. Indeed, multiple factors control sea-ice thickness (the forcing, the resolution, the physics, the assimilation technique, the data used for assimilation), and it is beyond the scope of this study to determine which factors dominate. In section 2, we introduce four sea-ice thickness data sets from different reanalyses as well as the respective data processing systems. We also introduce four kinds of reference data: two from satellite altimeters and two from in situ observations. In addition, we introduce sea-ice motion data set derived from satellites to help

investigate the seasonal variation and spatial distribution of sea-ice thickness. In section 3, we first compare all four reanalyses with ULS and ASPeCt records, and then we evaluate the spatial uncertainty of reanalyzed sea-ice thickness using ICESat-1 and Envisat observations. The seasonal variation and spatial distribution of sea-ice thickness differences between reanalyses and observations are also discussed. In section 4, we discuss the uncertainties and limitations of all reference data sets and summarize the conclusions.

**2 Data and methods**

Sea-ice thickness in the Weddell Sea from four reanalyses are evaluated against observations from satellite altimeters, moored upward looking sonars and ship observations. For comparison with Envisat, the modeled ice thickness data are gridded onto the Envisat product 50-km polar stereographic grid using linear interpolation. To enable comparison with ICESat-1 sea-ice





thickness estimates the reanalyses are gridded onto a 100 km Equal-Area Scalable Earth (EASE) grid (Brodzik et al., 2012),
also using linear interpolation. Comparisons are made using monthly means. Before comparing with in situ observations, such
as ULS and ASPeCt, all reanalyses and altimeter sea-ice thickness data are linearly interpolated to the locations of in situ
observations. In order to mitigate temporal gaps between the observations and reanalyses, the instantaneous ULS sea-ice
thickness data are monthly bin averaged before comparison. When comparisons are against monthly ASPeCt sea-ice thickness,
all available daily records around specified model grids are averaged monthly. However, the small temporal coverage of
ASPeCt impedes its representativeness, and the uncertainty of ASPeCt should be taken into consideration in the evaluation.
Besides, we exclude the IceBridge sea-ice thickness in our evaluation because the period of coincidence between this data set
and SOSE/NEMO-EnKF is very short.

## 2.1 Sea-ice thickness from four reanalyses

The German contribution of the Estimating the Circulation and Climate of the Ocean project Version 2 (GECCO2) is an ocean
synthesis based on MITgcm. GECCO2 assimilates abundant hydrographic observations by the adjoint method starting from
1948 (Köhl, 2015). This synthesis is only constrained by ocean measurements without any sea ice data assimilation.

Similar to GECCO2, the Southern Ocean State Estimate (SOSE) is also an ocean and sea ice estimate based on MITgcm using
the 4-D Var method (Mazloff et al., 2010). SOSE has been constrained by various kinds of observations, such as Argo and
CTD profiles, sea surface temperature and height from satellite observations, as well as mooring data. Also, SOSE assimilates
the satellite sea ice concentration data from the National Snow and Ice Data Center (NSIDC). SOSE has been widely used in
various studies (e.g., Abernathey et al., 2016; Cerovečki et al., 2019). In this paper, we evaluate the SOSE sea-ice thickness
provided from 2005 to 2010 at a resolution of 1/6° (Mazloff et al., 2010).

Massonnet et al. (2013) produced an Antarctic ice thickness reanalysis based on the Nucleus for European Modelling of the
Ocean (NEMO) ocean model coupled with the Louvain-la-Neuve sea Ice Model Version2 (LIM2) ice model, using the
ensemble Kalman Filter (EnKF), which is referred to NEMO-EnKF in the following text. Satellite sea ice concentration is
assimilated in this model by which the sea ice thickness is improved, exploiting the covariances between sea ice concentration
and sea ice thickness. The ice thickness in this data set has a spatial resolution of 2 degrees and has been used to investigate
the variability of salinity in the Southern Ocean (Haumann et al., 2016).

The Global Ice-Ocean Modeling and Assimilation System (GIOMAS) is based on the Parallel Ocean Model (POM) coupling
with a 12-category thickness and enthalpy distribution (TED) ice model (Zhang and Rothrock, 2003). The TED model
simulates sea ice ridging processes explicitly following Thorndike et al. (1975) and Hibler (1980). This data set includes
monthly ice thickness, concentration, growth/melt rate as well as ocean heat flux from 1970 to present. GIOMAS assimilates





sea ice concentration as described in Lindsay and Zhang (2006), and its ice thickness is evaluated to have good agreement with satellite observations in the Arctic.

## 2.2 Sea-ice thickness from altimeters

Currently, large-scale Antarctic ice thickness observations mainly come from laser and radar altimeters, among which the laser
altimetry data of Antarctic sea-ice thickness obtained from ICESat-1 are widely used due to its mature retrieval algorithm (Kurtz and Markus, 2012; Kern et al., 2016). Laser altimeters observe the total freeboard (the part above the sea level), and sea-ice thickness can be inferred from freeboard with different algorithms (Kurtz and Markus, 2012; Markus et al., 2011). Even though differences are found among these products, the spatial distribution of sea ice thickness generally shows similarities. We use a new ICESat-1 sea-ice thickness product retrieved from a modified ice density approximation suggested
by Worby, because this data was reported to have low biases relative to ASPeCt and may accurately reproduce seasonal thickness variations (Kern et al., 2016). Due to the extensive spatial coverage and relatively high accuracy of ICESat-1, we use this monthly mean sea-ice thickness product as a reference to evaluate the sea-ice thickness of the four reanalyses. Periods of availability of this product are given in Table 2. Though used as a reference, note that ICESat-1and ASPeCt data are biased low when compared to the ULS and Envisat data (Figure 3b).


Another large-scale sea-ice thickness data set used here are from the Sea Ice Climate Change Initiative (SICCI) project. SICCI includes Envisat (2002-2012) and CryoSat-2 (2010-2017) sea-ice thickness with spatial resolution of 50 km in the Antarctic (Hendricks et al., 2018). This new Antarctic sea-ice thickness dataset was published in August of 2018. Both Envisat and Cryosat-2 carrry a radar altimeter. The radar altimeter can measure the ice freeboard (total freeboard minus snow depth) instead
of only total freeboard as measured by ICESat-1, but with less accuracy. Previous studies indicated that Envisat overestimates the ice thickness because the radar signal can reflect inside the snow layer or even at the snow surface rather than reflect at the ice-snow interface (Willatt et al., 2010; Wang et al., 2020). The mean and modal sea-ice thickness from Envisat and Cryosat-2 are in good agreement during the sea-ice growth season, however, Envisat overestimates thin sea ice in the polynyas near the coasts and underestimates deformed thick ice in the multi-year sea ice region (Schwegmann et al., 2016). Due to the large
biases of Envisat sea-ice thickness, we only use these Envisat sea-ice thickness estimates as a supplement to ICEsat-1 when investigating the evolution of sea-ice thickness spatial distribution.

## 2.3 Sea-ice thickness from in situ measurements

The upward looking sonar (ULS) measures the draft (the underwater part of sea ice) continuously at a fixed location. In this paper, we use the sea-ice thickness from the upward looking sonar deployed in the Weddell Sea from 2002 to 2012. Ice draft
is converted into total ice thickness using the empirical relationship proposed by Harms et al. (2001), which is based on sea ice drilling measurements in the Weddell Sea, following Eq. (1):



$$D = 0.028+1.012d, \tag{1}$$

where $D$ represents total sea-ice thickness and $d$ represents the ice draft. The detailed processes of the sea-ice draft are described by Behrendt et al. (2013). This equation is skilful for thicknesses between 0.4 and 2.7 m, with a coefficient of determination
($r^2$) of 0.99, but overestimates thin ice with thicknesses less than 0.4 m. Even though the drilling cases included the snow layers, the empirical equation ignores the variations of snow depth. Owing largely to the sea-ice draft accuracy of 5 cm in the freezing/melting seasons and 12 cm in winter, the accuracy of the ULS sea-ice thickness is estimated to be 8 cm in freezing/melting seasons and 18 cm in winter.

Sea-ice thickness measurements based on the Antarctic Sea Ice Processes & Climate (ASPeCt) protocol are also used to evaluate the sea-ice thickness. The ASPeCt includes visual sea-ice thickness observations within 6 nautical miles of ship tracks with the period from 1981 to 2005. Errors in ice thickness are estimated to be ±20% of total thickness for level ice and ±30% for deformed ice thicker than 0.3 m. A simple function of undeformed sea-ice thickness, average sail height, and the fractional ridged area is used to compute the mean ice thickness (Worby et al., 2008). It is noted that the ASPeCt data tends to
underestimate mean sea-ice thickness because ships usually avoid thick sea ice.

**2.4 Sea-ice motion from satellite**

In order to attribute possible reasons for biases in sea-ice thickness, the sea-ice motion data set known as Polar Pathfinder Daily Sea Ice Motion Vector, version 4 from NSIDC is employed as a reference data (Tschudi et al., 2019). The daily sea-ice motion vectors are retrieved based on a block tracking method from sequential imagery using multiple sensors, including the
Scanning Multichannel Microwave Radiometer (SMMR), Special Sensor Microwave/Imager (SSM/I), Special Sensor Microwave Imager and Sounder (SSMIS), Advanced Microwave Scanning Radiometer-Earth Observing System (AMSR-E) and Advanced Very-High-Resolution Radiometer (AVHRR). In summer, when most sensors failed to retrieve ice motion, the ice motion vectors are mainly derived from wind speed estimates and buoy observations. The ice motion derived from multi-sources were merged using optimal interpolation (Isaaks and Srivastava, 1989). The monthly sea-ice motion vectors were
acquired from the daily ice motion vectors.

Based on the comparison with independent buoy observations, Schwegmann et al. (2011) indicated that NSIDC sea-ice motion vectors underestimate the meridional and zonal sea-ice velocity by 26.3% and 100%, respectively. Following Haumann et al. (2016), we use a simple correction for the NSIDC sea-ice motion vectors by multiplying the meridional speed by 1.357 and
the zonal speed by 2.000.



## 3 Results

### 3.1 Comparison with sea-ice thickness from upward looking sonars

In this section, we use sea-ice thickness from upward looking sonars to evaluate the abovementioned four reanalyses as well as other reference observations. Because thick deformed sea ice is found in the southern and western Weddell Sea (Behrendt
et al., 2013; Kurtz et al., 2012), the 13 ULS stations are divided into four sub-regions (Figure 1b): the Antarctic Peninsula (AP, including Station 206, 207 and 217), the central Weddell Sea (CWS, including Station 208, 209 and 210), the southern coast (SC, including Station 212, 232 and 233), and the eastern Weddell Sea (EWS, including Station 227, 229, 230 and 231). The classification criterion is based on the locations of ULS stations (Figure 1a) and long-term averaged ULS sea-ice thickness (Figure 1b). Under this classification, the Antarctic Peninsula is dominated by deformed thick sea ice, and the eastern Weddell
Sea by newly formed ice. The central Weddell Sea has both first-year ice and deformed sea ice, and the southern coast has both first-year ice and land-fast sea ice (Harms et al., 2001; Behrendt et al., 2013). The aggerate temporal span of ULS observations in AP, CWS, SC and EWS is 148, 79, 185 and 272 months, respectively.

Then we compare the ice thickness distribution from the reanalyses with ULS observations in 13 positions in the Weddell Sea
(Figure 2a). As presented in Table 1, SOSE has shorter period than the other three reanalyses. To include the most available data records in the inter-comparsion, the periods of GECCO2, NEMO-EnKF and GIOMAS are from 1990 to 2008, while the period of SOSE is from 2005 to 2008. The results indicate that for each data set, the most probable sea-ice thickness is less than 0.2 m. The NEMO-EnKF and ULS have local maxima in the distribution of 0.4-0.6 m. GIOMAS has local maxima at 1.2-1.4 m. Meanwhile, the probability density function (PDF) of GECCO2 and SOSE decreases with increasing sea-ice
thickness. None of the reanalyses have sea ice thicker than 2.2 m, though thicknesses of this magnitude are observed by ULS (Figure 2a).

The Taylor diagram (Figure 2b) indicates that the correlation coefficients (CCs) of all six data sets are larger than 0.4, and SOSE has the highest CC of 0.77. The maximum and minimum of root-mean-square error (RMSE) is 1.15 m of Envisat and
0.71 m for SOSE. The normalized standard deviations (STD) of sea-ice thickness from four reanalyses data sets are lower than 0.62, while the STD of Envisat and ICESat-1 are larger than 1.0. Compared with four reanalyses, ICESat-1 has higher STD that close to 1.0, which means ICESat-1 could reproduce the variation of sea-ice better than four reanalyses. It is noted that the relatively short ICESat-1 record (13 months) limits the accuracy of this assessment.

In the Antarctic Peninsula (Figure 3a), GECCO2, NEMO-EnKF and GIOMAS have CCs around 0.4, and SOSE has the highest CC of 0.62. All RMSEs for the four reanalyses are larger than 0.7 m. The STDs of the four reanalyses and Envisat are lower than the ULS. ICESat-1 has the largest CC of 0.74 and a STD of nearly 1.0. In the central Weddell Sea (Figure 3b), the CCs of the six data sets are all higher than 0.7. The STD of GECCO2, SOSE, NEMO-EnKF and GIOMASis 0.85, 0.52, 0.97 and



1.03, respectively. That means that GECCO2, NEMO-EnKF and GIOMAS could well reproduce the variation of the sea ice

thickness in the CWS. In addition, Envisat overestimates the interannual variability of sea-ice thickness significantly in this region as its STD is larger than 2.0. In the southern coast (Figure 3c), the CC of GECCO2, SOSE, NEMO-EnKF and GIOMAS is 0.50, 0.79, 0.50 and 0.52, respectively.  The normalized STD of GECCO2, SOSE, NEMO-EnKF and GIOMAS is 0.37, 0.53, 0.26 and 0.54, respectively, indicating that all reanalyses underestimate the sea-ice thickness variability, especially for the NEMO-EnKF. SOSE performs best among four reanalyses, with a high CC of 0.79 and a low RMSE of 0.66 m. In the eastern

Weddell Sea (Figure 3d), the CCs of GECCO2, SOSE, NEMO-EnKF and GIOMAS are 0.87, 0.90, 0.88 and 0.92, respectively. The normalized STD is 0.91, 0.76, 0.86 and 1.93, implying GECCO2, SOSE and NEMO-EnKF well reproduce the seasonal thickness variation of first-year ice. ICESat-1 has a lower CC (0.66) and STD (0.29), partly resulting from the large uncertainty of ICESat-1 ice thickness measuring the first-year ice thickness in this region, particularly in the summertime. Envisat has the lowest CC (-0.19) and highest RMSE (2.06 m) among all data sets, and its STD is comparable with GIOMAS.


SOSE has larger CCs than the other three reanalyses in the regions close to the coast (Antarctic Peninsula and Southern Coast). Even though SOSE uses the same MITgcm ice-ocean model as GECCO2, its higher spatial resolution of 1/6°resolves more small-scale dynamical processes in these regions. But in the regions with large amounts of newly formed ice (the central Weddell Sea and the eastern Weddell Sea), SOSE tends to underestimate sea-ice thickness with lower STDs than the other

reanalyses. GECCO2 and NEMO-EnKF have similar statistics in the four sub-regions. They perform best in the regions dominated by newly-formed ice (eastern Weddell Sea). GIOMAS has moderate performance in the regions close to the coast and performs best in the central Weddell Sea, with the highest CC of 0.92 and lowest RMSE of 0.40 m. GIOMAS shows excessive variability in the eastern Weddell Sea with a STD of 1.93.

**3.2 Comparisons with ice thickness from the ASPeCt**

The monthly sea-ice thickness distribution histograms (Figure 4a) show that the three reanalyses (GECCO2, NEMO-EnKF, GIOMAS) have distributions weighted more towards sea-ice thicknesses less than 1.8 m than ASPeCt does. We exclude SOSE in evaluation due to its relatively short period, because the ASPeCt observations used here are from 1981 to 2005, though there are extensive ASPeCt observations from 2005 to 2012, but the sample records are very limited in the Weddell Sea. While there

are little sea-ice thicknesses greater than 1.8 m in GECCO2, NEMO-EnKF and GIOMAS, ASPeCt has recorded ice thicker than 3.0 m. Given that the ASPeCt observations from an area with a 6 nautical mile radius (~11.1 km) are compared with models with ~ 60 km spatial resolution, this is unsurprising. The ship observations show the pack ice to be a highly varied and complicated mixture of different ice types. The concentration, thickness, and topography may vary significantly over a short spatial distance. Compared with ASPeCt, GECCO2 has more sea ice with thickness ranging from 0.5 m to 1.25 m, and

GIOMAS has more sea ice with thickness ranging from 1.3 m to 1.8 m. NEMO-EnKF mainly overestimates sea-ice thickness within the bins from 0 to 1.0 m. In addition, the sea-ice thickness of GECCO2, NEMO-EnKF and GIOMAS seem to





concentrate within the range of 0.8 to 1.4 m, 1.1 to 1.7 m, and 1.1 to 1.7 m, respectively (Figure not shown). These thicknesses are mainly found over the first-year sea-ice area of the eastern Weddell Sea and ice edge (Figure 4b-d). In these regions, reanalyses tend to overestimate sea-ice thickness with respect to ASPeCt, which is consistent with the results reported in

Timmermann et al. (2004). The small-scale spatial and temporal variation of ice thickness, which is represented in the ASPeCt observations, is not captured by the reanalyses.

### 3.3 Comparison with sea-ice thickness from ICESat-1

In this section, we compare sea-ice thickness from four reanalyses (GECCO2, SOSE, NEMO-EnKF and GIOMAS) with that from ICESat-1 for the period from 2005 to 2008. Based on the statistics of aggregate sea-ice thickness, all four reanalyses

underestimate ice thickness close to 1 m (Table 3). The root-mean-square error (RMSE) of the four reanalyses exceed 0.6 m, and the maximum and minimum is 0.8 m (GIOMAS) and 0.6 m (SOSE), respectively. The correlations between the four reanalyses and ICESat-1 are low, and the maximum correlation coefficient is only 0.31 (NEMO-EnKF). It should be noted that the ICESat-1 records are very limited, only in October, November, February, March, May and June (see Table 2 for more information). Following Kern and Spreen (2015) and Kern et al. (2016), when comparing with ICESat-1, we use October and

November to represent spring (hereafter Spring-ON) February and March to represent autumn (hereafter Autumn-FM), May and June to represent winter (hereafter Winter-MJ), respectively. Based on the interannual variation of ice thickness distribution (ITD) from Autumn-FM to Spring-ON (Figure 5), we find that ICESat-1 thickness is much thicker than that of the reanalyses except GIOMAS in Spring-ON. The ITD of ICESat-1 shows peaks mainly around 1.2 m (ice thickness < 0.5 m are truncated) while the four reanalyses have peaks in the low sea-ice thickness bins (<1.0 m) and very little ice thicker than 2.0

m. The modal sea-ice thickness of ICESat-1 has weak interannual variation in different seasons (red dots in Figure 5), but the modal sea-ice thickness of NEMO-EnKF and GIOMAS have significant interannual variation in Autumn-FM. In addition, the modal and mean ice thickness of ICESat-1 have significant seasonal variation (e.g., modal thickness decreases from 1.7 m to 0.9 m from austral Autumn-FM to Winter-MJ due to the new ice formation, and increases to 1.3 m from Winter-MJ to Spring-ON due to the thermodynamic and dynamic processes) and the variability of mean ice thickness is less than that of modal ice

thickness (Table 4). Compared with ICESat-1, only NEMO-EnKF has a similar variation of modal ice thickness from Autumn-FM to Spring-ON, while GECCO2, SOSE and GIOMAS have monotonically increasing modal ice thickness. For mean sea-ice thickness, all four reanalyses have similar variability and the GIOMAS has the smallest variation.

In addition to the aggerate sea-ice thickness statistics, the spatial difference of thickness between four analyses and ICESat-1

is also investigated. The ICESat-1 data show that ice thicker than 2.5 m, mainly located in the western Weddell sea and with a location shifting from the southwestern Weddell Sea in Autumn-FM to the northwestern Weddell Sea in Spring-ON (Figure 6). In Autumn-FM, all reanalyses underestimate ice thickness. For GECCO2 and SOSE, negative biases up to 1.5 m almost cover the entire Weddell Sea and the negative biases of NEMO-EnKF and GIOMAS mainly occur in the area near the coast. Considering the ICESat-1 thickness may be biased low (Kern et al., 2016), this suggests that these reanalyses may not well



represent coastal processes. The spatial averaged differences between models and ICESat-1 are -1.30 m (GECCO2), -0.63 m (NEMO-EnKF) and -0.75 m (GIOMAS), respectively. In Winter-MJ, all reanalyses still underestimate sea-ice thickness along the Antarctic Peninsula (AP) and in the western Weddell Sea, and GIOMAS overestimates thickness in the central Weddell Sea and near the Ronne Ice Shelf of the southern Weddell Sea, where new sea ice is found. All four reanalyses underestimate sea-ice thickness by up to 1.5 m in the north edge of sea-ice cover. In Spring-ON, the area of thickness underestimation of all

four analyses shrinks to the western Weddell Sea along the AP and the northern edge of ice cover, while a slight overestimation is also found in the central and eastern Weddell Sea. In addition, GIOMAS overestimates ice thickness near the Ronne Ice Shelf in the southern Weddell Sea, which is thought to be an important source of new sea ice (Drucker et al., 2011). The overestimation is likely due partially to GIOMAS's explicit simulation of sea ice ridging processes, which tends to create thick ridges. It may also be due to the generally low ICESat-1 thickness values when compared to ULS and Envisat data (see Figure

3d above).

### 3.4 Comparison with seasonal evolution of sea-ice thickness from Envisat

The comparison with ICESat-1 thickness in section 3.3 is limited by the temporal coverage of ICESat-1, in particular, the seasonal evolution cannot be fully quantified. Although the Envisat sea-ice thickness is with larger biases than ICESat-1 thickness (Schwegmann et al., 2016; Wang et al., 2020), it is still can be applied in comparing the seasonal evolution of the

sea ice thickness, because the Envisat ice thickness covers all the seasons, and its spatial distribution has a good spatial correlation with the ICESat-1 (figure not shown here).

In this section, based on the Envisat sea ice thickness data, we focus on the comparison of seasonal variation of the spatial distribution of sea-ice thickness averaged from 2005 to 2008. Following with the seasonal classification in Holland and Kwok

(2012), the summer, autumn, winter and spring hereinafter refer to January to March, April to June, July to September and October to December, respectively. The spatial distribution of sea ice thickness of NEMO-EnKF shows the most similarity with Envisat over the year (Figure 7). GECCO2 and SOSE have similar sea-ice thickness distributions all year round, while GECCO2 is much thicker. The thickest ice of GECCO2 and SOSE is mainly located in the southern Weddell Sea and southwestern Weddell Sea, respectively. NEMO-EnKF reproduces the thick sea ice (>1.5m) over the region in the

northwestern Weddell Sea from winter to spring. Compared with other models, GIOMAS has the largest amount of thick ice (>2.0 m), and it is mostly located in the western and southern Weddell Sea and occurs in all seasons. In addition, different from other data sets, GIOMAS has a large area of sea ice thicker than 1.5 m between -25°W and 0°E over the eastern Weddell Sea from autumn to spring.

The sea-ice concentration is also analyzed, as it is closely tied to sea-ice thickness via dynamics and thermodynamics. Benefiting from data assimilation approaches, all models have a similar spatial distribution of sea-ice concentration with respect to satellite observations (Figure 8). GECCO2, which has not assimilated sea-ice concentration, has a high concentration





in the southern Weddell Sea, while the other three models have the high concentrations found mostly in the southwestern Weddell Sea. It is worth noting that the SOSE sea ice concentration shows a "river" pattern with relatively low sea ice concentration around the primary meridian in autumn and winter. This phenomenon can be attributed to the open-ocean polynya in 2005 and has also been reported by Abernathey et al. (2016).

Since sea-ice thickness distributions are associated with ice advection, we investigate the sea ice motion effects on the spatial distribution of sea ice thickness. Because Envisat does not measure ice motion, therefore, the satellite ice motion data from the National Snow and Ice Data Center are used instead (Tschudi et al., 2019). In addition, we also calculate the divergence of ice motion to investigate the influence of ice motion on the variation of sea-ice thickness. As shown in Figure 7, a clockwise ice motion is the leading pattern in the Weddell Sea, known as Weddell Gyre, especially in wintertime. GECCO2 has weak ice motion and weak convergence in the southern Weddell Sea (the cyan rectangle in Figure 8), while the other three reanalyses show apparent westward ice motion. That gives rise to less ice accumulation along the AP in GECCO2. In addition, the westward movement of the SOSE, NEMO-EnKF and NSIDC ice velocity fields with ice convergence in the southwestern Weddell Sea are in favour of the dynamic thickening. Compared to NEMO-EnKF and GIOMAS in summer through autumn, SOSE has a stronger sea ice circulation advecting more ice toward the north-western Weddell Sea and the coast of the AP. SOSE has rapid ice motion for all seasons, especially near the Antarctic Peninsula in the western Weddell Sea and the coast near Queen Maud Land (QMD) in the southern Weddell Sea. The high ice speed of SOSE in this region may result from its relatively thin sea ice. Based on the satellite data, the convergence is mainly in the middle and eastern Weddell Sea. The divergence is mainly in the southern and western Weddell Sea, which are regions of new sea-ice formation and sea-ice deformation, respectively (Figure 8). GECCO2 mainly has convergence in all seasons. The strong divergence and convergence of SOSE alternatively occur in the south-eastern Weddell Sea and the northern edge of the sea-ice cover. The sea ice motion convergence of NEMO-EnKF is relatively weak, but widespread, and is generally consistent with satellite inferences. GIOMAS shows an abnormal divergence in the eastern Weddell Sea in autumn, which may result from its thick ice in this region diagnosed in section 3.3.

In order to quantitatively estimate the influence of sea ice advection on thickness in the southwestern Weddell Sea, we calculate sea ice flux across two sections. The zonal section (from 70°W to 25°W, 65°S) captures outflow from the western Weddell Sea (Harms et al., 2001). Flux across the meridional section (65°S to 72°S, 25°W) is also diagnosed to form a closure (Figure 8, blue and red line). Here, we use sea-ice area flux instead of the volume flux to exclude the thickness influence. All models underestimate the sea-ice area flux across 25°W, especially for GECCO2 and GIOMAS (Figure 10a). The ice area flux in GIOMAS is approximately half of that in NSIDC (Table 5). In the 65°S section, GIOMAS has smaller northward ice area flux, which favors thick ice staying in the southwestern Weddell Sea. With respect to the NSIDC product, GECCO2 and SOSE have relatively small ice inflow in the 25°W section ($0.95\times10^3$km$^2$/mon and $0.30\times10^3$km$^2$/mon) and relatively high outflow in the 65°S section ($3.06\times10^3$km$^2$/mon and $3.13\times10^3$km$^2$/mon), which favors thin ice in the southwestern Weddell Sea. SOSE and





NEMO-EnKF have similar ice flux in the 25°W section, but NEMO-EnKF has better ice thickness distribution than SOSE according to Figure 7. NEMO-EnKF has smaller ice flux in the 65°S section and a better correlation with NSIDC. We find that accurate northward ice motion in the western Weddell Sea is related to thick ice accumulation in the southwestern Weddell Sea and actual sea-ice thickness distribution in the Weddell Sea.

## 4 Discussion and summary

In this paper, we evaluate sea-ice thickness in the Weddell Sea from four reanalyses against observations from altimeters, mooring and visual observations. It should be noted that although this evaluation is based on most of the available observations in the Antarctic Weddell Sea, there are still some uncertainties and limitations in this evaluation. For example, due to the temporal coverage of the reanalyses and reference data, the large-scale evaluation against ICESat-1 and Envisat are limited to 2005 to 2008, and mainly focuses on the seasonal evolution and spatial distribution of ice thickness. The evaluation against ASPeCt is from 1981 to 2005. Furthermore, Schwegmann et al. (2016) already showed that Envisat sea-ice thickness underestimates thick ice and overestimates thin ice compared to CryoSat-2, in addition, the Envisat sea-ice thickness has different interannual variability compared with the in situ ULS observations. Nevertheless, the Envisat thickness has been still used to investigate the seasonal evolution of sea ice in this study. These limitations could be further addressed when more ice thickness observations are available in the future.

It comes to a conclusion that current sea ice-ocean models reproduce sea ice thickness in the Weddell Sea with a varying degree of realism. Compared with ASPeCt, GECCO2, NEMO-EnKF and GIOMAS have deficiencies reproducing the small spatio-temporal variation of thickness in regions dominated by first-year ice. Compared with ICESat-1 and ULS sea-ice thickness, all four reanalyses underestimate ice thickness in the western and north-western Weddell Sea with highly deformed sea ice (mean ice thickness > 1.5 m) from Autumn-FM to Spring-ON. To be particular, GIOMAS and SOSE ice thickness performs best in the Central and the South Coast of the Weddell Sea, respectively, while GECCO2 and NEMO-EnKF could reproduce new ice evolution in the eastern Weddell Sea. GIOMAS tends to overestimate first-year ice thickness in the eastern Weddell Sea, especially in spring. Besides the explicit simulation of ice ridging, the convergence of GIOMAS sea ice in the central Weddell Sea may be an important cause of the positive bias in sea-ice for this data. Compared with Envisat, only NEMO-EnKF did well reproducing the clock-wise shift of thick ice from the western Weddell Sea in winter to the north-western Weddell Sea in spring. Our study also indicates the northward ice motion in the western Weddell Sea along the Antarctic Peninsula has an important influence on ice thickness distribution in the Weddell Sea.

This study shows, to accurately infer the variability of the Antarctic sea-ice volume (not only the Weddell Sea) in the context of global climate change, there is still room to further improve the Antarctic sea-ice reanalyses and prediction, and the possible ways include improve the ice-ocean model physics via optimizing model parameters (e.g., Sumata et al., 2019), and assimilate



the observed ice-ocean observations (in particular the satellite derived sea ice thickness) with an ensemble based sea ice-ocean

multi-variate data assimilation approach (e.g., Mu et al., 2020).

*Data availability.* The GECCO2 sea-ice thickness are available at https://icdc.cen.uni-hamburg.de/1/daten/reanalysis-ocean/gecco2.html (Köhl, 2015). The SOSE sea-ice thickness are available at http://sose.ucsd.edu/sose_stateestimation_data_05to10.html (Mazloff et al., 2010). The NEMO-EnKF sea-ice thickness are

available at http://www.climate.be/seaice-reanalysis (Massonnet et al., 2013). The GIOMAS sea-ice thickness are available at http://psc.apl.washington.edu/zhang/Global_seaice/data.html (Zhang and Rothrick, 2003). The ICESat-1 sea ice thickness are available at http://icdc.cen.uni-hamburg.de/1/projekte/esa-cci-sea-ice-ecv0/esa-cci-data-access-form-antarctic-sea-ice-thickness.html (Kern et al., 2016). The CryoSat-2 and Envisat sea ice thickness are available at https://dx.doi.org/10.5285/b1f1ac03077b4aa784c5a413a2210bf5 (Hendricks et al., 2018). The ASPeCt sea ice thickness are

available at http://aspect.antarctica.gov.au/data (Worby et al., 2008). Sea ice velocity are available at https://nsidc.org/data/NSIDC-0116/versions/4 (Tschudi et al., 2019). The Weddell Sea upward looking sonar ice draft are available at https://doi.pangaea.de/10.1594/PANGAEA.785565 (Behrendt et al., 2013).

*Author contributions.* QY and LM developed the concept of the paper. QS analyzed all the data and wrote the manuscript. All

authors assisted during the writing process and critically discussed the contents.

*Competing interests.* The authors declare that they have no conflict of interest.

*Acknowledgments.* The authors would like to thank Jinlun Zhang from University of Washington for his invaluable advice in

improving the manuscript. This is a contribution to the Year of Polar Prediction (YOPP), a flagship activity of the Polar Prediction Project (PPP), initiated by the World Weather Research Programme (WWRP) of the World Meteorological Organisation (WMO). We acknowledge the WMO WWRP for its role in coordinating this international research activity. This study is supported by the National Natural Science Foundation of China (No. 41941009, 41922044), the Guangdong Basic and Applied Basic Research Foundation  (No. 2020B1515020025), and the Fundamental Research Funds for the Central

Universities (No. 19lgzd07). The authors would like to thank European Space Agency (ESA) for providing the Envisat and ICESat-1 data, and Alfred Wegener Institute, Helmholtz Centre for Polar and Marine Research (AWI) for providing Weddell Sea ULS data. François Massonnet is a F.R.S.-FNRS Research Fellow.




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



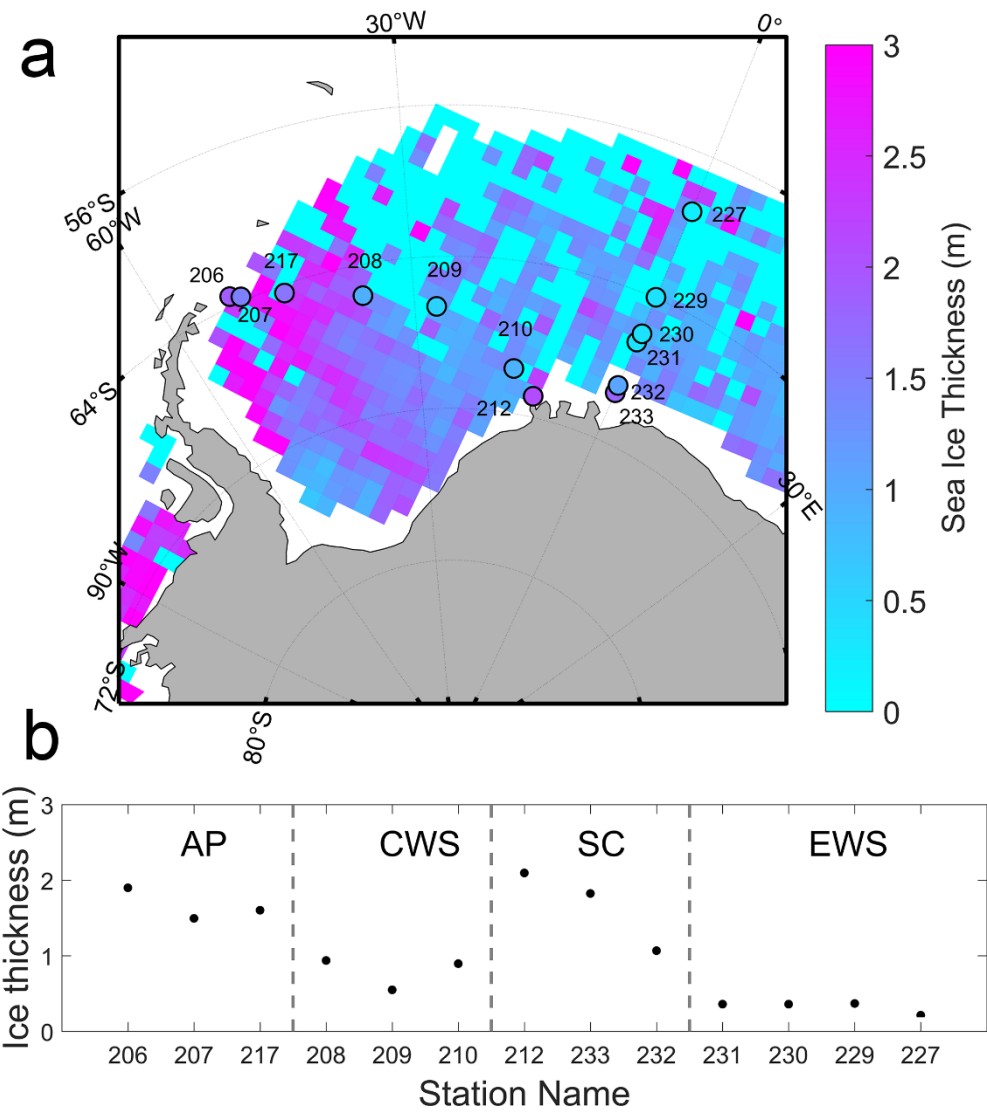

**Figure 1: a) The locations of the Weddell Sea moored upward looking sonar stations, with their long term mean thicknesses coloured.**
**The ICESat-1 sea-ice thickness in autumn of 2005 is coloured in the background. b) The mean ULS sea-ice thickness from west to the east in the Weddell Sea. Grey dotted lines divide the 13 stations into four parts: the Antarctic Peninsula (AP), the central Weddell Sea (CWS), the southern coast (SC), and the eastern Weddell Sea (EWS).**

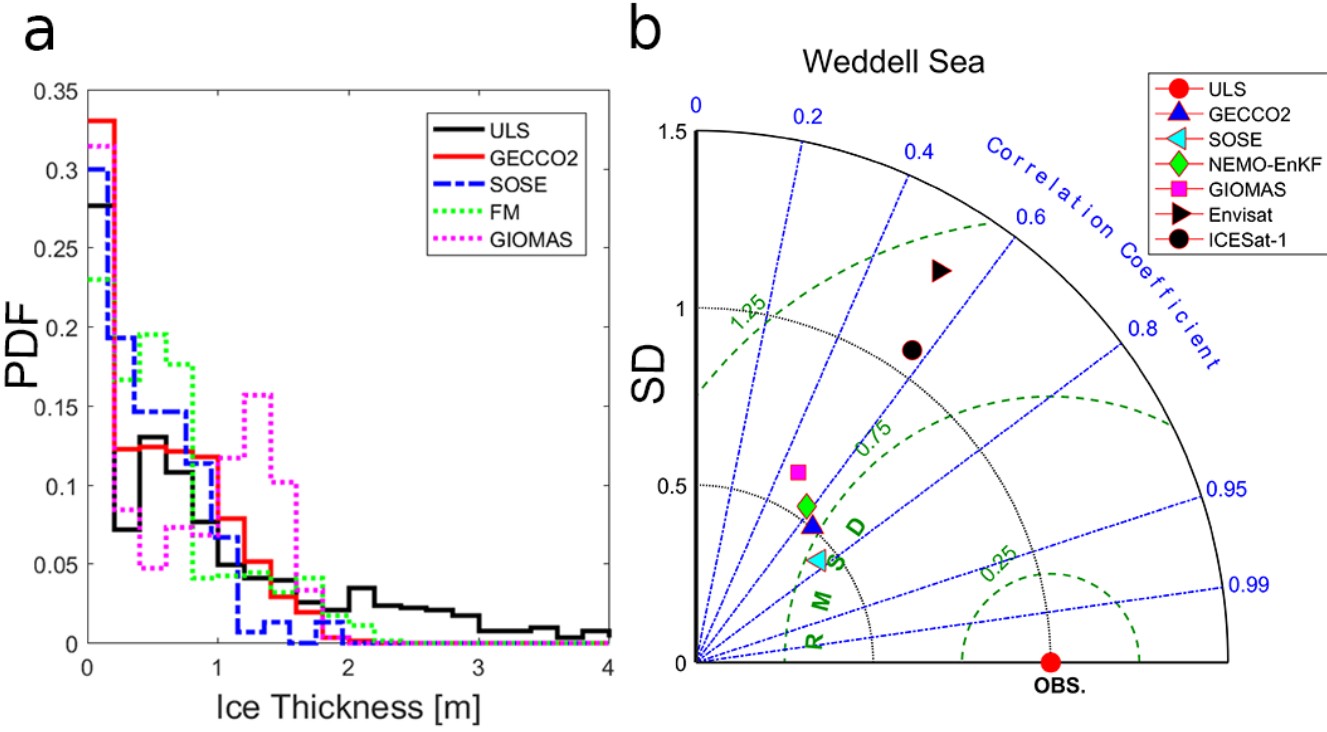


**Figure 2: a) Probability density distributions (PDF) of monthly sea-ice thickness from ULS and four reanalyses at the 13 ULS locations of the Weddell Sea. b) Normalized Taylor diagram for monthly sea ice thickness of four reanalyses as well as Envisat and ICESat-1 with respect to the sea-ice thickness from upward-looking sonar from 1990 to 2008 in the Weddell Sea. The green dashed lines indicate the normalized root-mean-square error (RMSE).**







Figure 3: Same as Figure 1b, but for the four sub-regions: a) Antarctic Peninsula, b) central Weddell Sea, c) southern coast, and d) eastern Weddell Sea.



Figure 4: a) Histograms of sea-ice thickness from ASPeCt and three reanalyses. Locations of model sea-ice thickness are shown in b) GECCO2 for a range of 0.8 to 1.4 m, c) NEMO-EnKF for a range of 1.1 to 1.7 m, d) and GIOMAS for a range of 1.1 to 1.7 m.


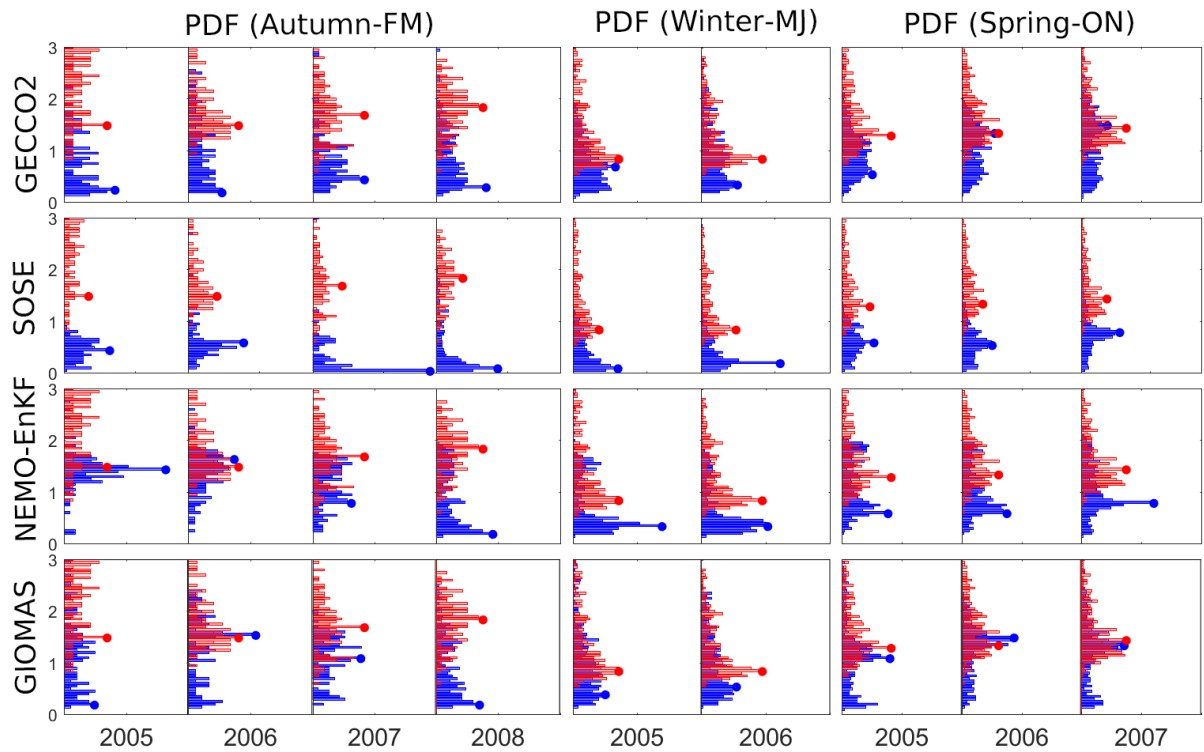

**Figure 5: The variation of monthly ice thickness distribution from reanalyses (blue) and ICESat-1 (red) in Autumn-FM (left), Winter-MJ (middle) and Spring-ON (right). The blue and red dots represent the modal ice thickness of reanalyses (blue) and ICESat-1, respectively.**






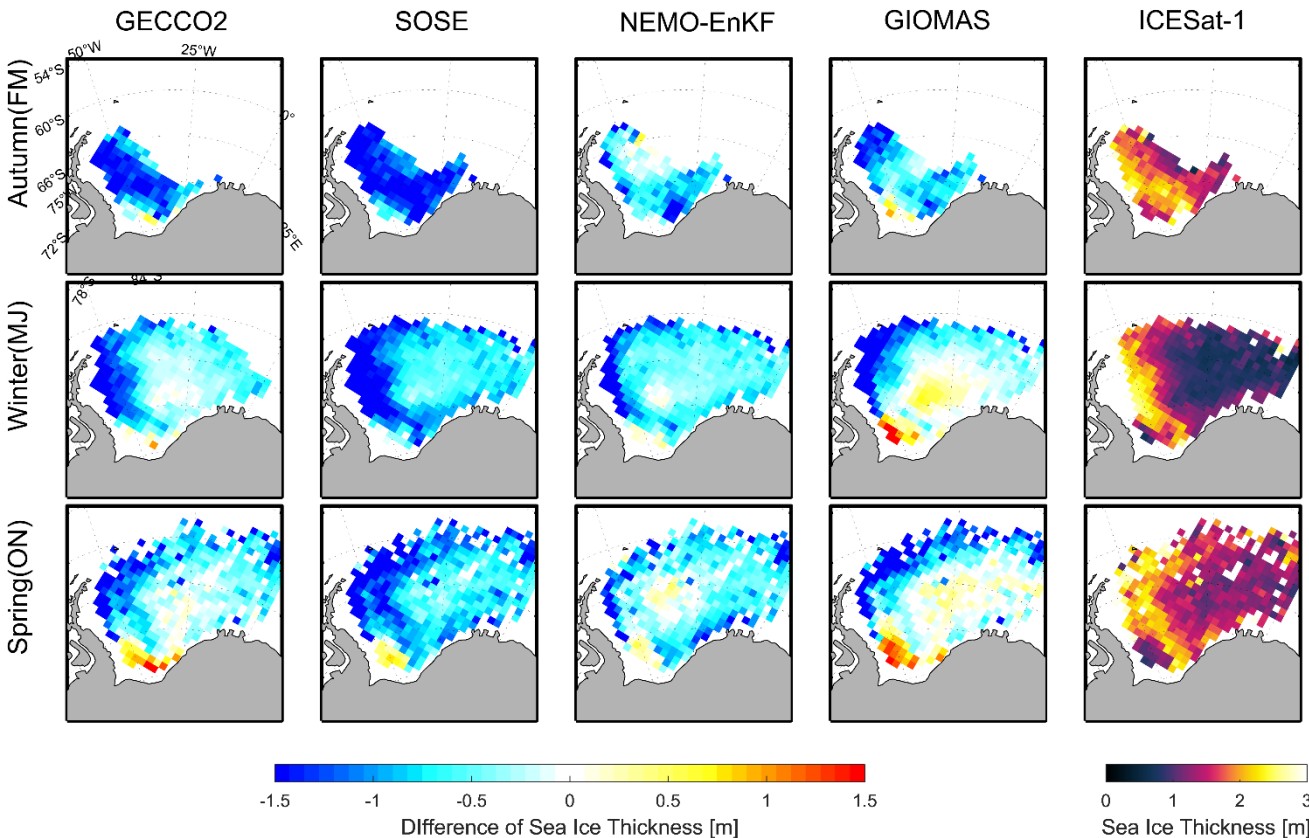

**Figure 6: The Differences of sea ice thickness between GECCO2 (first column), SOSE (second column), NEMO-EnKF (third column), and GIOMAS (fourth column) and ICESat-1 in Autumn-FM (first row), Winter-MJ (second row) and Spring-ON (third row).The contours in last column represent the autumn sea-ice thickness of ICESat-1.**




Figure 7: Seasonal mean sea-ice thickness (summer to spring) from four reanalyses and Envisat for the 4-yr period 2005-2008.







**Figure 8: Seasonal mean sea ice concentration (summer to spring) for the 4-yr period 2005-2008. The overlapped vectors represent sea-ice velocity from respective data sets.**



**Figure 9: Same as Figure 9 but for divergence of sea ice motion.**





**Figure 10: a) Monthly sea-ice area fluxed westward into the southwestern Weddell Sea across the 25°W section and b) fluxed northward out of the southwestern Weddell Sea across the 65°S section from 2005-2008.**



Table 1: Introduction of the four reanalyses data systems used in this study.

|  | GECCO2 | SOSE | NEMO-EnKF | GIOMAS |
|---|---|---|---|---|
| period | 1948.01-2016.12 | 2005.01-2010.12 | 1979.01-2009.11 | 1979.01-present |
| domain | Global | South Hemisphere | Global | Global |
| Spatial resolution | 1°×1/3° | 1/6°×1/6° | 2°×2° | 4/5°×4/5° |
| Vertical levels | 50 z-levels | 42/52 z-levels | 31 z-levels | 25 z-levels |
| Ocean model | MITgcm | MITgcm | NEMO (Madec, 2008) | POP |
| Ice model | MITgcm embedded sea-ice model (Zhang and Hibler, 1997; Hibler, 1980) | Same as GECCO2 | LIM2 (Fichefet and Maqueda, 1997; Timmerman et al., 2005) | TED (Zhang and Rothrock, 2003) |
| Assimilation method for ocean | 4-D Var (adjoint) method | 4-D Var (adjoint) method | / | / |
| Assimilation method for sea ice concentration | / | 4-D Var (adjoint) method | Ensemble Kalman filter (Mathiot et al., 2012) | Nudging(Lindsay and Zhang, 2006) |
| Sea ice concentration used for assimilation | / | NSIDC (25 km×25 km) | EUMETSAT-OSISAF (12.5 km×12.5 km) | HadISST (1°×1°) |
| Atmospheric forcing | NCEP-NCAR daily reanalysis (Kalnay et al., 1996) | adjusted NCEP/ adjusted ERA-interim | NCEP-NCAR daily reanalysis (Kalnay et al., 1996) | NCEP-NCAR daily reanalysis (Kalnay et al., 1996) |





Table 2: ICESat-1 measurement periods in this study. Abbreviations given in parentheses in each cell are used throughout the paper to denote the respective period. Spring-ON refers October and November, Autumn-FM refers February and March, Winter-MJ refers May and June, respectively.

| Year | Winter-MJ | Autumn-FM | Spring-ON |
|------|-----------|-----------|-----------|
| 2004 | 18 May-21 June (MJ04) | 17 February-21 March (FM04) | 3 October-8 November (ON04) |
| 2005 | 20 May-23 June (MJ05) | 17 February-24 March (FM05) | 21 October-24 November (ON05) |
| 2006 | 24 May-26 June (MJ06) | 22 February-27 March (FM06) | 25 October-27 November (ON06) |
| 2007 | Winter-MJ | 12 March-14 April (MA07) | 2 October-5 November (ON07) |
| 2008 | - | 17 February-21 March (FM08) | - |

Table 3: The mean ice thickness bias, root-mean-square error estimate and correlation between ICESat-1 and four sea-ice thickness reanalyses.

| Reanalysis | Mean error (m) | RMSE (m) | Correlation |
|------------|----------------|----------|-------------|
| GECCO2 | -1.02 | 0.71 | 0.18 |
| SOSE | -1.20 | 0.63 | 0.20 |
| NEMO-EnKF | -0.99 | 0.68 | 0.31 |
| GIOMAS | -0.90 | 0.79 | 0.17 |

Table 4: The modal and mean sea-ice thickness of ICESat-1 and four reanalyses from 2005 to 2008.

| | Autumn-FM | | Winter-MJ | | Spring-ON | |
|------------|-----------|----------|-----------|---------|-----------|----------|
| | Modal (m) | Mean (m) | Modal (m) | Mean (m) | Modal (m) | Mean (m) |
| ICESat-1 | 1.70 | 1.92 | 0.85 | 1.36 | 1.30 | 1.65 |
| GECCO2 | 0.30 | 0.89 | 0.35 | 0.47 | 0.55 | 1.04 |
| SOSE | 0.10 | 0.45 | 0.20 | 0.37 | 0.60 | 0.65 |
| NEMO-EnKF | 1.45 | 1.08 | 0.35 | 0.68 | 0.75 | 1.03 |
| GIOMAS | 0.20 | 1.02 | 0.40 | 0.95 | 1.35 | 1.16 |

650



655

Table 5: Mean sea-ice volume flux biases, root-mean-square error and correlation through the 25°W and 65°S sections between four reanalyses and satellite observations. (Unit: 103km2/mon, positive/ negative sign means the outflow and inflow into region outlined by red and blue lines in Figure 10)

| | Section 25°W | | | | Section 65°S | | | |
|---|---|---|---|---|---|---|---|---|
| | Net flux | Bias | RMSE | Correlation | Net flux | Bias | RMSE | Correlation |
| GECCO2 | 0.95 | 1.57 | 1.46 | 0.67 | -3.06 | -0.22 | 1.41 | 0.86 |
| SOSE | 0.30 | 0.92 | 1.04 | 0.85 | -3.13 | -0.29 | 2.04 | 0.68 |
| NEMO-EnKF | 0.49 | 1.12 | 1.07 | 0.84 | -2.53 | 0.49 | 1.62 | 0.84 |
| GIOMAS | 1.28 | 1.91 | 1.40 | 0.75 | -0.50 | 2.34 | 1.64 | 0.81 |