# Peer review of "Evaluation of Sea-Ice Thickness from four reanalyses in the Antarctic Weddell Sea"

_The Cryosphere, 2020_

## Referee Comment (RC1) · Keguang Wang (Referee) · 2 Jun 2020

In this study, Shi et al. evaluate sea ice thickness (SIT) in the Weddell Sea from four reanalyses of coupled ocean and sea ice models, against two in-situ observations and two remote sensing datasets. Their results show that these reanalyses have limited success compared with the observations, and they stress the importance of sea ice motion and deformation on the SIT simulations. Modeling and observations of the Antarctic SIT, even for a single region such as the Weddell Sea, are very challenging. This study provides useful insights not only in the SIT reanalyses but also in the SIT observations, which will be a benefit to the sea ice research community. The Introduction, and Data and methods are well written. However, the Results part needs significant improvement. I suggest the manuscript accepted after major revision.

[Figure]

General Comments:

1. There is inconsistency during the comparison in terms of the data. In the "Data and methods" part, the authors state "Comparison are made using monthly means", however, when in 3.3 Comparison with sea-ice thickness from ICESat-1, they are using seasonal mean. This inconsistency must be fixed. It will be much better that the authors describe how they make the comparison in the exact sections.

2. Section 3.1. It remains unclear what kind of mooring data are using here. According to the statement "The aggregate temporal span of ULS observations in AP, CWS, SC and EWS is 148, 79, 185 and 272 months", and consider the numbers of the mooring in these regions, there should be large difference in the mooring data regarding the time duration. I suggest the authors add a plot in their Figure 1 showing the temporal evolution of the mooring observed SIT that are actually used in their comparisons.

3. Figure 4. Not sure why the authors use SITs from different locations in the three re-analyses (Figs. 4b-d). This means they also use different ASPeCT SIT when comparing with the different reanalyses. What can we infer from such different comparisons? I suggest the authors use a consistent comparison: Use the same ASPeCT SIT, with reanalysis SITs interpolated to the same time and same location.

4. Section 3.3. It is not clear what kind of manipulations here for the reanalysis SIT. The authors state "we use October and November to represent spring . . .". However, according to Table 2, the ICESat-1 measurement is irregular, and no full month measurements. Do the authors use the same dates as the observations, or just simply use the full two-month reanalysis data? As the authors here try to compare the mean, it is very important to compare the exact corresponding data in terms of time and locations. Also the authors need give a test with confidence level for the comparison.

5. Line 280-281. "Compared with ICESat-1, only NEMO-EnKF has a similar variation of modal ice thickness from Autumn-FM to Spring-ON, while GECCO2, SOSE and GIOMAS have monotonically increasing model ice thickness". It seems to me 2005 &

2006 for SOSE, and 2006 for GIOMAS have similar seasonal variations in Figure 5. Table 4 looks somewhat misleading as its modal SITs not necessary in the same year.

Specific Comments:

1. Line 40. Not clear. Rephrase it.

2. Line 62. "... it is also reported to have uncertainties due to ...". All observations have uncertainties. Perhaps more important to note what kind of uncertainty it is.

3. Line 104. "all available daily records around specified model grids are averaged monthly". How far away from that model grid?

4. Lines 110 & 114. I think the adjoint method and 4-D Var method are the same here. Can the authors give a brief of their differences?

5. Line 195. "Figure 1b", better remove "b".

6. Line 217. "It is noted that the relatively short ICESat-1 record (13 months) limits the accuracy of this assessment". Not sure how the authors arrive to this statement.

7. Line 233. "Envisat has the lowest CC (-0.19) and highest RMSE (2.06 m) among all data sets, and its STD is comparable with GIOMAS". These numbers are almost unbelievable. The authors have any explanation of this? Is there any reference supporting similar findings? Or is there some error in the manipulation of the data?

8. Line 238. "But in the regions with large amounts of newly formed ice (the central Weddell Sea and the eastern Weddell Sea), SOSE tends to underestimate sea-ice thickness with lower STDs than the other reanalyses". It looks to me no data here to support that SOSE underestimates SIT.

9. Line 331. "Figure 7" should be "Figure 8"?

---

## Referee Comment (RC2) · Céline Heuzé (Referee) · 26 Jun 2020

I would like to start by pointing out that I was asked to serve as a reviewer in June.

This manuscript evaluates the Southern Ocean sea ice thickness produced by four reanalyses against observations from AUVs, ships and satellites. The manuscript is quite good honestly. Sure it is not how I would have written it, so that I regularly took note of "[whatever] is missing" that I erased after reading the information a few lines later, but nothing that really impairs understanding. There's a lot of figures, but they all have a reason to be here.

I have two somewhat methodological points that I would like to see addressed, and a series of comments to improve the readability, but I consider that it should not require

a lot of work. Hence my evaluation "minor revisions".

1) The regions On Fig 1, you present the four regions into which you split the Weddell Sea, and that you analyse in Fig 3. You base that split on data from ULS, but you present only their mean, not the uncertainty attached to it. I am particularly surprised that 210 and 212 would be in different regions. So at least on Fig 1b, add the errors bars. Then modify the region split if needed.

2) The more recent time period and long term perspective Most of the analysis is performed on the time period common to all four reanalyses (late 2000s), which I understand. Unfortunately, it is a bit old and short. Southern Ocean sea ice has behaved very differently since. So please, include a short extra subsection dedicated to comparing GIOMAS and GECCO to SICCI (CryoSat2 at least) and APP. Ideally, also add something about trends in these reanalyses.

Now for the more minor comments, in order of appearance:

Line 109: say that all the information to come is summarised in Table 1. Try to write this entire section in a more structured manner, giving the same information about all four products (at least time period and resolution).

Line 140: you mention ASPeCT now, but only introduce the product line 170.

Fig 1a: add the lines separating the four regions plotted there too Fig 1b: see comment above, add the error bars, and potentially modify your region division accordingly.

Fig 3d: why is the correlation negative for Envisat? What happens? Is the bias mostly in summer or winter?

Fig 4b-d: why are you showing different thickness bands for different products? They are not even the thicknesses you comment on in the text. Please show only one range, so that the reader can compare the reanalyses.

Table 3: have you checked whether the reanalyses are correlated with each other? It

is suspicious that they all seem to have similar biases when compared against ICESat.

Fig 7: present it like Fig 6, as difference against reference rather than actual values. This way, we can compare with Fig 6 (alternatively, present Fig 6 like Fig 7).

Fig 8 (and text corresponding): since the sea ice concentration is about right, and that all reanalyses present similar biases in thickness when compared to satellite retrievals, can it be that the thickness retrievals are the ones that are not perfect yet? Sea ice concentration retrieval is after all more mature.

Line 331: you meant to refer to Fig 8 here.

Line 345/Fig 9: I know you write that you will not investigate the reasons for biases in the reanalyses, but I find the north sea ice of GIOMAS in winter/spring surprising. Is the reanalysis known for having too fast an Antarctic Circumpolar Current? Or is the ice too thin/mobile?

Line 342-346: you forgot to refer to Fig 9 here. The caption of Fig 9 refers to itself instead of Fig 8 by the way.

Table 5: the units need to be fixed. Indicate the net flux in the reference product as well (at least in the caption).

Line 373: not "sea ice ocean models", reanalyses. Sea ice ocean models have their own series of problems, but that's beyond the scope of this review.

---

## Referee Comment (RC3) · Daniel Price (Referee) · 15 Jul 2020

Review of The Cryosphere Discussion submission tc-2020-71 'Evaluation of Sea-Ice Thickness from four reanalyses in the Antarctic Weddell Sea'

GENERAL COMMENTS

The accurate large-scale measurement and reporting of trends in Antarctic sea ice thickness are two of the major challenges for contemporary geophysical science. Once developed, giving these trends context amongst the multiple drivers of Antarctic sea ice volume change, from natural and anthropogenic forcing over multi-decadal timescales will present and even greater challenge. Given their limited temporal resolution, this cannot be achieved using observational datasets alone. This manuscript evaluates

sea-ice thickness from four reanalyses against observational data in the Weddell Sea sector of the Southern Ocean. Reanalyses can play an important role in providing context for change during the observational period, and this work has impressively pulled together a large amount of data, from a variety of sources to provide an evaluation.

It is interesting to see these results and impressive that monthly sea ice thickness distribution is generally represented as expected in this sea ice sector. The simplification of the real world in these reanalyses is obviously a major concern particularly regarding their resolution, this is not a critique of this work, but a description of the current state of the science. The major limitation appears to be that these reanalyses are struggling to capture the thicker end of the sea ice distribution in the Weddell Sea (and from other reading, the entire Southern Ocean) and therefore omitting an important segment of the statistical information on the sea ice thickness distribution. This also highlights the fact that they are failing to, and often provide no attempt to simulate sea ice dynamics, a process at the core of Antarctic sea ice production. Their sparse resolution and missing physics related to smaller scale deformation processes is a serious limitation. There are clearly major developments to be made before these reanalyses datasets can be used to inform the community about sea ice thickness trends in the Southern Ocean. This work successfully highlights this current limitation.

I find the manuscript meets the set criteria for publication in The Cryosphere but I do have some concern related to the 'Originality' and 'Significance' criteria. Although the particular area of study and reanalysis models are on the whole original, when placed in a wider context, other work has already visited this question (Uotila et al. (2019) is an intercomparison while Massonet et al. (2013); Holland et al. (2014) are reanalysis assessments also comparing results to observational data). It is difficult to gauge how valuable continued comparisons are, especially related to the difference/variability of the physical processes that construct sea ice thickness in the models. For instance, have significant advancements been made in the reanalyses evaluated here, since the publication of other studies to warrant new evaluations? I have no specific recommendation for this. This is perhaps beyond the scope of the expected review process though I feel it is important to highlight it.

With the consideration of this concern above left to the discretion of the editor, I support the publication of the manuscript if the specific points outlined below are incorporated into the work. I would like to place specific emphasis on the tightening up of the results section and an attempt to provide more quantitative conclusions.

SPECIFIC COMMENTS

1. Readers may benefit from a concise explanation of the general principles of sea ice thickness estimates from reanalysis products. It will be difficult for non-experts in this specific field to grasp the processes considered and limitations during the construction of a sea ice thickness reanalysis product. I appreciate this is the point of the references but it is often helpful to provide an insight as part of the text to assist the reader (supported by references).

2. I appreciate it is sometimes difficult to fit all the relevant information into the limited word count of an abstract, but I think the reader (and work) would benefit from some sort of quantified reporting in the abstract. Terms like 'well reproduce' are somewhat subjective, is there a way to effectively provide a quantitative measure in the abstract of how well these reanalysis perform compared to one another and the observations? i.e. report the key results in a quantitative manner. This could in some way be related to a 'score' suggested in point 5 below.

3. Although the manuscript is written well and results are well displayed, it would be useful to maintain the colour coding of each reanalysis/observational datasets throughout the figures to avoid confusion.

4. It is clear that the reanalyses underestimate the sea ice thickness distribution when compared to ICESat-1 but maybe some attention should be given to the description of this comparison in the results. "ICESat-1 thickness is much thicker than that of
the reanalyses except GIOMAS in Spring-ON" – is this entirely accurate? GECCO2 also has two instances where the modes are similar when compared to ICESat-1, in Spring-ON (2007) the mode is higher in GECCO2, Spring-ON (2006) it appears to be the same. NEMO-EnkF also has two examples in Autumn-FM where the modes are higher (2006) and similar (2005). Is GIOMAS in Spring-ON really that notable? I am a little confused by the plots in Figure 5 from visual inspection - why does the ICESat-1 thickness distribution change in the same season and year for the SOSE comparison? i.e. the ICESat-1 distributions seem to stay the same in the PDFs in the same year/season for the other plots but the distribution is shifted in the SOSE plots. Is this to do with some different sampling from different geographical coverage of the reanalyses? In addition, why is 2007/2008 omitted for Winter-MJ and 2008 for Spring-ON? Was this decision driven by a lack of ICESat-1 data for comparison? It is stated in the text (L264) that the 'we compare sea-ice thickness from four renanalyses...with that from ICESat-1 for the period from 2005 to 2008' but this does not appear to be the case in the corresponding figure. In Table 2 the ICESat-1 measurement periods are described and 2007 (Winter-MJ) does not have a '-' indicating the data is absent but instead 'Winter-MJ' is written. Also 2004 is shown but does not appear to be part of the described analysis, is there a reason for this? My main point related to this section of the study is that there seem to be some discrepancies in how the data is described and how it is presented in figures. This needs to be looked at and all data and descriptions must be consistent.

5. I would expect that the community would look to evaluations like this to understand what reanalyses could be useful for supporting their own work. As the manuscript currently reads, it is difficult to digest and really understand the limitations of each of the reanalyses. It may provide some clarity and assist the readers understanding of the results to have a table with the key parameters the authors are trying to evaluate (including but not limited to thickness, relationship between mean/mode, min/max thickness accuracy, spatial accuracy, sea ice growth/seasonal evolution of thickness, open ocean vs. coastal regimes, ice motion –divergence/convergence) and a score

evaluating how well they have performed. This is not an explicit request, but just a suggestion for the authors to consider in order to improve the communication of important information from this work.

TECHNICAL COMMENTS

The temporal span of this investigation isn't immediately clear. Please make clear by including the time frame of the inter-comparison in the abstract and introduction (the analysis overlaps are written at the beginning of Section 4). I understand it is intermittent given the different lengths of the observational datasets and different reanalyses but some indication via a well worded summary, early in the manuscript would be useful.

L28 – 'crucial component of the Earth system', understand what authors mean but perhaps more specific 'climate system' for example.

L89 – add 'a' between 'introduce' and 'sea-ice'.

Section 2.1 ∼ L110 and L130 - should a spatial resolution be reported for GECCO2/GIOMAS as is given for the other reanalyses? I see they are in Table 1 but why report some resolutions in the text and not others?

L118 and L124 – 'âĄř' used in one instance and 'degrees' in another, perhaps adopt one standard.

L136 – to be absolutely accurate perhaps reword '(the part above the sea level)' to '(combined ice and snow height above local sea level)'.

L140 – 'suggested by Worby'? Is a complete reference available?

L149 – I understand that the limitations of radar altimeters are not the focus of this study but the complexity of the technique/its limitations in the Antarctic are understated by these few sentences. Perhaps include reference to other studies highlighting this to provide the reader with some context if they require it. This takes me to another point,

it doesn't appear CS-2 is used in the analysis, why is it described in the data section?

L149 – More accurate to say 'the radar altimeter is expected to' (and then provide relevant references) as opposed to 'the radar altimeter can measure'.

L159 – Use acronym 'ULS' once it is provided and throughout manuscript use acronyms/abbreviations once they are supplied e.g. L199 'Antarctic Peninsula' to 'AP' as it is shortened on L195.

L164 – I don't think 'skilful' is appropriate here, perhaps 'accurate' or 'approximates thickness well' or something similar.

L167 – What are these uncertainties? 5 cm/8cm/18 cm etc? Are they a spread around the mean +/- 5 cm or direct positive deviations from other reference measurements? If so are there references for these expected accuracies?

Figure 1 – Standard deviation is abbreviated to SD in the figure but to STD in the text, these should be consistent.

Figure 3 caption – Capitalise 'southern coast'.

Figure 5 – Thickness (m) is not actually labelled on the y-axis. Insert '(red)' after second mention of ICESat-1 in the caption.

L290 – 'this means that the reanalyses may not well represent coastal processes' – what do the authors specifically mean here in reference to sea ice? Dynamics and convergence against the coast? Interaction with the coastline? Inaccurate bathymetry or coastal currents? Some of the concluding statements are a little vague. I think the study would benefit from being more specific and shed light on the limitations that need to be addressed.

L290 – Why is SOSE not included in the spatially averaged differences here?

L325 – 'primary' to 'prime'.

[Figure]

L362 – insert 'satellite' before 'altimeters'.

L369 – 'still' before 'been'.

L388 – 'improve' to 'improving'.

L388 – 'assimilate' to 'assimilating'.

Figure 6 – What time period is this data comparison for? Are they seasonal averages for all years?

In acknowledgements – ICESat-1 data is provided by NASA and NSIDC not ESA.

REFERENCED EXISTING REANALYSES

Holland, P. R., N. Bruneau, C. Enright, M. Losch, N. T. Kurtz, and R. Kwok, 2014: Modeled Trends in Antarctic Sea Ice Thickness. J. Climate, 27, 3784–3801, https://doi.org/10.1175/JCLI-D-13-00301.1.

Massonnet, F., Mathiot, P., Fichefet, T., Goosse, H., König Beatty, C., Vancoppenolle, M., and Lavergne, T.: A model reconstruction of the Antarctic sea ice thickness and volume changes over 1980–2008 using data assimilation, Ocean Modelling, 64, 67-75, 10.1016/j.ocemod.2013.01.003, 2013.

Uotila, P., Goosse, H., Haines, K. et al. An assessment of ten ocean reanalyses in the polar regions. Clim Dyn 52, 1613–1650 (2019). https://doi.org/10.1007/s00382-018-4242-z.

---

## Author Comment (AC1) · 14 Sep 2020

**Responses to referee #1**

Dear Dr. Keguang Wang:

Thanks for your helpful comments to improve this manuscript.

Below, we repeat each comment and insert our replies in the text. All responses are in blue font for clarity of reading.

Qinghua Yang

On behalf of all the authors

**Main Comments:**

**Point 1:** There is inconsistency during the comparison in terms of the data. In the "Data and methods" part, the authors state "Comparison are made using monthly means", however, when in 3.3 Comparison with sea-ice thickness from ICESat-1, they are using seasonal mean. This inconsistency must be fixed. It will be much better that the authors describe how they make the comparison in the exact sections.

*Response:* In the old version, we compared four reanalyzed SIT products with ULS, ASPeCt by monthly mean, but with seasonal mean for products from Envisat and ICESat-1. Now, we deleted "Comparisons are made using monthly means" in the "Data and methods" part, then described the related information in individual sections instead. For example, we added "All ULS recorded once a second are averaged into monthly ice draft" in Line 194 in section 3.1 before "Because thick…"

**Point 2:** Section 3.1. It remains unclear what kind of mooring data are using here. According to the statement "The aggregate temporal span of ULS observations in AP, CWS, SC and EWS is 148, 79, 185 and 272 months", and consider the numbers of the mooring in these regions, there should be large difference in the mooring data regarding the time duration. I suggest the authors add a plot in their Figure 1 showing the temporal evolution of the mooring observed SIT that are actually used in their comparisons.

*Response:* The previous description indeed could make the reader confused about the ULS data used in this section. Actually, we collected all daily ULS records of 13 stations in the Weddell Sea from 1990 to 2008. Due to the different ice conditions during this period, the duration of records in the four different sectors (AP, CWS, SC and EWS) are quite different. The aggregate time span is $40(206) + 84(207) + 24(217) = 148$ months in AP, $40(208) + 10(209) + 23(210) = 73$ months in CWS, $23(212) + 37(233) + 125(232) = 185$ months in SC, and $91(231) + 28(230) + 108(229) + 45(227) = 272$ months in EWS. That is to say, the correct aggregate time span of ULS observations in AP, CWS, SC and EWS is 148, 73, 185 and 272 months, respectively. CWS has the fewest observations because it is far away from the coast and has a relatively long ice-covered time. Though CWS has fewer observations than AP and SC, its standard deviation (SD) is lower than AP and SC (new Figure 1b). Considering average of sea ice thickness as well as their SDs, we think current division is reasonable.

According to your suggestion, we added the time series of 15-day moving average sea ice thickness based on daily records of all 13 stations. Besides, we added the standard deviations of daily sea ice thickness as error bars in all mean ice thickness (new Figure 1b) to represent the variations of all stations.

[Figure]

New Figure 1: a) The ICESat-1 sea-ice thickness in autumn of 2005 in the Weddell Sea and the locations of the moored upward looking sonars with their mean thicknesses shaded. b) The mean ULS sea-ice thickness from west to the east in the Weddell Sea. The error bars represent the standard deviation of daily ice thickness for individual stations. Grey dotted lines divide the 13 stations into four parts: the Antarctic Peninsula (AP), the central Weddell Sea (CWS), the southern coast (SC), and the eastern Weddell Sea (EWS). C) The time series of daily sea ice thickness of all 13 stations after a 15-day moving average.

**Point 3:** Figure 4. Not sure why the authors use SITs from different locations in the three reanalyses (Figs. 4b-d). This means they also use different ASPeCT SIT when comparing with the different reanalyses. What can we infer from such different comparisons? I suggest the authors use a consistent comparison: Use the same ASPeCT SIT, with reanalysis SITs interpolated to the same time and same location.

*Response:* We are sorry because the old caption for Figure 4 was wrong and this caused the misleading. It is not "Locations of model sea-ice thickness are shown in b ...", but should be "Locations of modal sea-ice thickness are shown in b ...". We have corrected this.

**Point 4:** Section 3.3. It is not clear what kind of manipulations here for the reanalysis SIT. The authors state "we use October and November to represent spring : : :". However, according to Table 2, the ICESat-1 measurement is irregular, and no full month measurements. Do the authors use the same dates as the observations, or just simply use the full two-month reanalysis data? As the authors here try to compare the mean, it is very important to compare the exact corresponding data in terms of time and locations. Also the authors need give a test with confidence level for the comparison.

*Response:* We used a two-month mean SIT for reanalyses in Section 3.3 when comparing with ICESat-1. Considering the irregular months of ICESat-1, we performed a time-weighted calculation for all four reanalyses in the new comparison. For example, if the temporal span of FM04 is from 17 January to 21 March, which includes 13 days in February and 21 days in March, then all SIT reanalyses will be averaged by $(13/34)*SIT_{Feb}+(21/34)*SIT_{Mar}$. That is what we plotted for the new Figures 5 & 6. The main change in Figure 5 is for GIOMAS reanalysis. The difference between GIOMAS modal SIT and ICESat-1 modal SIT decreased in FM05, FM07, MJ05 and MJ06 after using the new monthly averaged SIT. The spatial pattern of monthly SIT in the new Figure 6 is generally consistent with that in the old Figure 6. The main difference occurs in the GIOMAS winter, where the new figure 6 has less area with differences around -0.5 m over the central Weddell Sea. Besides, all reanalyses SIT are biased to ICESat-1 SIT with a t-test.

[Figure]

Old Figure 5: The variation of monthly ice thickness distribution from reanalyses (blue) and

ICESat-1 (red) in Autumn-FM (left), Winter-MJ (middle) and Spring-ON (right). The blue and red dots represent the modal ice thickness of reanalyses (blue) and ICESat-1, respectively.

[Figure]

New Figure 5: The variation of monthly ice thickness distribution from GECCO2 (blue), SOSE (cyan), NEMO-EnKF (green), GIOMAS (pink) and ICESat-1 (red) in Autumn-FM (left), Winter-MJ (middle) and Spring-ON (right). The colored dots represent the modal ice thickness. In order to make the histogram plots readable, different reanalyses has different x range.

[Figure]

Old Figure 6: The Differences of sea ice thickness between GECCO2 (first column), SOSE

(second column), NEMO-EnKF (third column), and GIOMAS (fourth column) and ICESat-1 in Autumn-FM (first row), Winter-MJ (second row) and Spring-ON (third row).The contours in last column represent the autumn sea-ice thickness of ICESat-1.

[Figure]

New Figure 6: The Differences of sea ice thickness between GECCO2 (first column), SOSE (second column), NEMO-EnKF (third column), and GIOMAS (fourth column) and ICESat-1 in Autumn-FM (first row), Winter-MJ (second row) and Spring-ON (third row).The contours in last column represent the autumn sea-ice thickness of ICESat-1.

**Point 5:** Line 280-281. "Compared with ICESat-1, only NEMO-EnKF has a similar variation of modal ice thickness from Autumn-FM to Spring-ON, while GECCO2, SOSE and GIOMAS have monotonically increasing model ice thickness". It seems to me 2005 &2006 for SOSE, and 2006 for GIOMAS have similar seasonal variations in Figure 5. Table 4 looks somewhat misleading as its modal SITs not necessary in the same year.

*Response:* In the new version, the sentences starting from Line 279, "and the variability of mean ice thickness is less than that of modal ice thickness (Table 4)… " were deleted. Following your suggestion, we added "In most cases, modal ice thickness of reanalyses are lower than that of ICESat-1. For example, in 2008 Autumn-FM, four reanalyses have modal ice thickness lower than 0.3 m, indicating the newly formed sea ice. However, ICESat-1's modal ice thickness is around 1.5 m. SOSE and NEMO-EnKF have a similar variation of modal ice thickness from Autumn-FM to Spring-ON as ICESat-1 in 2005 and 2006. GIOMAS has a similar seasonal variation in 2005. GECCO2 fails to reproduce the decrease of modal ice thickness from Autumn-FM to Winter-MJ. This is because GECCO2 loses most thick ice in summer and thus has lower modal ice thickness than the other data sets." In addition, we deleted Table 4 in the new version to avoid misleading.

---

## Author Comment (AC2) · 14 Sep 2020

**Responses to referee #2**

Dear Dr. Céline Heuzé:

Thanks for your helpful comments to improve this manuscript.

Below, we repeat each comment and insert our replies in the text. All responses are in blue font for clarity of reading.

Qinghua Yang

On behalf of all the authors

**Main Comments:**

**Point 1:** The regions On Fig 1, you present the four regions into which you split the Weddell Sea, and that you analyse in Fig 3. You base that split on data from ULS, but you present only their mean, not the uncertainty attached to it. I am particularly surprised that 210 and 212 would be in different regions. So at least on Fig 1b, add the errors bars. Then modify the region split if needed.

*Response:* We divided 13 ULS stations according to their ice conditions. Following your suggestion, we added the uncertainty of daily SIT records (here we use the standard deviation representing the uncertainty) in the new figure 1b. The mean SIT of station 210 is similar to other stations of the CWS group, lower than that of the SC group. In addition, the uncertainties of station 208, 209 and 210 are all around 0.5 m, while the uncertainties of stations of SC group are all higher than 0.85 m. Therefore, we think our division criterion is reasonable. In order to provide more information on ULS records following another reviewer, we added the time series of daily SIT of all 13 stations after a 15-day moving average in the new Figure 1c.

[Figure]

New Figure 1: a) The ICESat-1 sea-ice thickness in autumn of 2005 in the Weddell Sea and the locations of the moored upward looking sonars with their mean thicknesses shaded. b) The mean ULS sea-ice thickness from west to the east in the Weddell Sea. The error bars represent the standard deviation of daily ice thickness for individual stations. Grey dotted lines divide the 13 stations into four parts: the Antarctic Peninsula (AP), the central Weddell Sea (CWS), the southern coast (SC), and the eastern Weddell Sea (EWS). C) The time series of daily sea ice thickness of all 13 stations after a 15-day moving average.

**Point 2:** The more recent time period and long term perspective most of the analysis is performed on the time period common to all four reanalyses (late 2000s), which I understand. Unfortunately, it is a bit old and short. Southern Ocean sea ice has behaved very differently since. So please, include a short extra subsection dedicated to comparing GIOMAS and GECCO to SICCI (CryoSat2 at least) and APP. Ideally, also add something about trends in these reanalyses

*Response:* Thanks for the suggestion. We added the comparison of GECCO2, GIOMAS, Cryosat-2 and APP-x from 2011 to 2016 in the Figure supp1. The SIT of GECCO2 and GIOMAS have an obvious lower SIT than SICCI (Cryosat-2), but their spatial pattern are similar to SICCI (Cryosat-2) in the period from 2005 to 2008. The thickest ice of GECCO2 and GIOMAS concentrate on the southwestern Weddell Sea in all seasons, without spatial shift as Cryosat-2 or Envisat shows. APP-x has lower SIT than Cryosat-2 in summer and autumn. Then, its SIT increases rapidly in winter and spring and has the thickest mean ice thickness in the spring of all four data sets. Its abnormal thick ice (> 3 m) in the central and eastern Weddell is contrary to the ULS observations in the literature. Besides, the spatial pattern of APP-x SIT presents obvious zonal symmetry. That is to say, the evolution of APP-x SIT is mainly controlled by thermodynamic processes and cannot well reflect the dynamic processes.

Though, we showed the yearly-mean sea ice thickness in the Weddell Sea of GECCO2, GIOMAS and APP-x from 2000 to 2019 (Figure supp2). The Cryosat-2 sea ice thickness has also been added to this plot. Even though the magnitudes of SIT are quite different among the four data sets, all of them show relatively high mean SIT in 2014. GECCO2 and GIOMAS present an upward trend from 2000 to 2019 (exceed 95% significance level), while APP-x presents a downward trend at the same time but cannot pass significance text. Currently, we still cannot conclude whether the upward signal of GECCO2 and GIOMAS SIT is realistic. Further, more reliable and longer data sets are needed to investigate the trend of Antarctic SIT is necessary.

We have not put this part in the main document, since both the CryoSat-2 and the APP-x ice thickness are with large uncertainties. In particular, the uncertainties of the radar altimeter can result from the inaccuracy snow-ice interface and snow-ice formation (Willatt et al., 2010), and also the surface type mixing and surface roughness (Schwegmann et al., 2016; Paul et al., 2018; Tilling et al., 2019).". APP-x almost cannot grasp the dynamical thickening of sea ice, that is to say, it cannot keep the memory of sea ice cover in the Weddell Sea, which exist abundant multi-year ice.

[Figure]

Figure supp1: Seasonal mean sea-ice thickness (summer to spring) from GECCO2, GIOMAS, SICCI and APP-x for the 6-yr period 2011-2016.

[Figure]

Figure supp2: Trends of yearly-mean sea ice thickness from 2000 to 2016. The dotted lines represent the linear regression fittings.

**Minor Comments:**

**1 Line 109:** say that all the information to come is summarised in Table 1. Try to write this entire

section in a more structured manner, giving the same information about all four products (at least time period and resolution).

*Response:* We added the following sentences at the end of Line 111 "Its horizontal spatial resolution is $1° \times 1/6°$.". At the end of Line 134, we added "The horizontal spatial resolution of GIOMAS is $4/5° \times 4/5°$.".

**2 Line 140:** you mention ASPeCT now, but only introduce the product line 170

*Response:* We changed the "ASPeCt" here as "ship-based observations".

**3 Fig 1a:** add the lines separating the four regions plotted there too Fig 1b: see comment above, add the error bars, and potentially modify your region division accordingly

*Response:* We added error bar in the new Figure 1b and explained the reason for using such division criterion in the response of Point 1.

**4 Fig 4b-d:** why are you showing different thickness bands for different products? They are not even the thicknesses you comment on in the text. Please show only one range, so that the reader can compare the reanalyses.

*Response:* First, we are sorry because the old caption for Figure 4 was wrong and this caused the misleading. It is not "Locations of model sea-ice thickness are shown in b …", but should be "Locations of modal sea-ice thickness are shown in b …". We have corrected this. Second, the different modal ice thickness corresponds to the leading ice types of different reanalyses. Their similar spatial locations mean the modal ice thickness is a representative parameter in comparison with ASPeCt SIT.

**5 Table 3:** have you checked whether the reanalyses are correlated with each other? It is suspicious that they all seem to have similar biases when compared against ICESat-1.

*Response:* We are very sorry as there is indeed something wrong with the old Table 3, please see the following new Table 3 (the old values are in parentheses). Based on the new results, NEMO-EnKF has obvious higher CC than the other three reanalyses and the CC of GIOMAS close to 0.

New Table 3: The mean ice thickness bias, root-mean-square error estimate and correlation between ICESat-1 and four sea-ice thickness reanalyses.

| Reanalysis | Mean error (m) | RMSE (m) | Correlation |
|---|---|---|---|
| GECCO2 | -0.67(-1.02) | 0.55(0.71) | 0.19(0.18) |
| SOSE | -0.99(-1.20) | 0.51(0.63) | 0.26(0.20) |
| NEMO-EnKF | -0.63(-0.99) | 0.44(0.68) | 0.54(0.31) |
| GIOMAS | -0.52(-0.90) | 0.68(0.79) | 0.03(0.17) |

**6 Fig 7:** present it like Fig 6, as difference against reference rather than actual values. This way, we can compare with Fig 6 (alternatively, present Fig 6 like Fig 7).

*Response:* Corrected. Please see the new Figure 7.

[Figure]

Figure 7: Same as Figure 6 but with respect to Envisat (last column) for the 4-yr period 2005-2008.

**7 Fig 8** (and text corresponding): since the sea ice concentration is about right, and that all reanalyses present similar biases in thickness when compared to satellite retrievals, can it be that the thickness retrievals are the ones that are not perfect yet? Sea ice concentration retrieval is after all more mature.

*Response:* Yes, the performance of sea ice concentration is better than the sea ice thickness, both for remote sensing retrieval and numerical modeling. Comparing with the sea ice concentration, both the satellite and the model ice thickness estimates have large errors, because the modeled ice concentration can be constrained by the satellite observations, but this is not the case for the modeled ice thickness.

**8 Line 331:** you meant to refer to Fig 8 here.

*Response:* Yes. We corrected this.

**9 Line 345/Fig 9:** I know you write that you will not investigate the reasons for biases in the reanalyses, but I find the north sea ice of GIOMAS in winter/spring surprising. Is the reanalysis known for having too fast an Antarctic Circumpolar Current? Or is the ice too thin/mobile?

*Response:* Good suggestion. On the one hand, the relative low ice concentration in the marginal ice zone and high ice motion speed of GIOMAS (Figure 8) than the other four data sets can accelerate the sea ice loss (local melting or northward advection). On the other hand, the thin ice thickness will make ice more mobile driven by the same wind speed. Currently, we still cannot give the explicit casualty between thin ice and fast ice motion of GIOMAS.

**10 Line 342-346:** you forgot to refer to Fig 9 here. The caption of Fig 9 refers to itself

instead of Fig 8 by the way

*Response:* Corrected.

**11 Table 5:** the units need to be fixed. Indicate the net flux in the reference product as well (at least in the caption).

*Response:* Corrected.

**12 Line 373:** not "sea ice ocean models", reanalyses. Sea ice ocean models have their own series of problems, but that's beyond the scope of this review.

*Response:* This sentence will be replaced by "We conclude that sea ice thickness reanalyses in the Weddell Sea have a varying degree of realism."

---

## Author Comment (AC3) · 14 Sep 2020

**Responses to referee #3**

Dear Dr. Daniel Price:

Thanks for your helpful comments to improve this manuscript.

Below, we repeat each comment and insert our replies in the text. All responses are in blue font for clarity of reading.

Qinghua Yang

On behalf of all the authors

**Main Comments:**

**Point 1:** Readers may benefit from a concise explanation of the general principles of sea ice thickness estimates from reanalysis products. It will be difficult for non-experts in this specific field to grasp the processes considered and limitations during the construction of a sea ice thickness reanalysis product. I appreciate this is the point of the references but it is often helpful to provide an insight as part of the text to assist the reader (supported by references)

*Response:* Good suggestion. We added some description:

Sea ice thickness is a prognostic variable in all ocean—sea ice models used to generate the reanalyses considered in this study. The use of a data assimilation scheme offers the possibility to provide revised estimates of sea ice thickness, by constraining the simulated model output with observations (ocean or sea ice, e.g., Sakov et al., 2012; Köhl, 2015; Mu et al., 2018).

Added reference:

Köhl, A., 2015: Evaluation of the GECCO2 ocean synthesis: transports of volume, heat and freshwater in the Atlantic. Q. J. Roy. Meteor. Soc., 141, 166-181.

Mu, L., M. Losch, Q. Yang, R. Ricker, S. N. Losa, and L. Nerger, 2018: Arctic-wide sea ice thickness estimates from combining satellite remote sensing data and a dynamic ice-ocean model with data assimilation during the CryoSat-2 period. J. Geophys. Res.-Oceans, 123, 7763-7780.

Sakov, P., F. Counillon, L. Bertino, K. A. Lisaeter, P. R. Oke, and A. Korablev, 2012: TOPAZ4: An ocean-sea ice data assimilation system for the North Atlantic and Arctic. Ocean Sci., 8, 633-656.

**Point 2:** I appreciate it is sometimes difficult to fit all the relevant information into the limited word count of an abstract, but I think the reader (and work) would benefit from some sort of quantified reporting in the abstract. Terms like 'well reproduce' are somewhat subjective, is there a way to effectively provide a quantitative measure in the abstract of how well these reanalysis perform compared to one another and the observations? i.e. report the key results in a quantitative manner. This could in some way be related to a 'score' suggested in point 5 below

*Response:* Good suggestion. A quantitatively description of the comparison results are necessary to know the performance of all four reanalyses. After a serious thinking, we tend to present values of root-mean-square error (RMSE) and correlation coefficient (CC), which are more objective instead of "score" ranking. It is noted that the CC with ULS means the temporal correlation between four reanalyses and ULS, while the CC with ICESat-1 means the spatial correlation because they are calculated by yearly mean SIT fields. Our results (Table 5) show that the SOSE has the highest CC of 0.77 and lowest RMSE of 0.72 m, when compared with

ULS ice thickness. All RMSEs are less than 0.9 m and all CCs are more than 0.4. Compared with ICESat-1, NEMO-EnKF has the highest CC of 0.54 and lowest RMSE of 0.44 m. CCs of the other three reanalyses are less than 0.3 and GIOMAS almost no spatial relation with ICESat-1.

By the way, we will add the RMSE anc CC results in the new abstract. But we cannot conclude the performance of reanalyses only by their RMSEs or CCs. First, the time coverage of ICESat-1 is quite limited. Second, the spatial representation of ULS data sets is very sparse.

Table 5. Statistics of four reanalyses with respect to ULS and ICESat-1.

|  | GECCO2 | SOSE | NEMO-EnKF | GIOMAS |
|---|---|---|---|---|
| ULS(RMSE) | 0.77 | 0.72 | 0.82 | 0.89 |
| ULS(CC) | 0.65 | 0.77 | 0.58 | 0.47 |
|  |  |  |  |  |
| ICESat-1(RMSE) | 0.55 | 0.51 | 0.44 | 0.47 |
| ICESat-1(CC) | 0.19 | 0.26 | 0.54 | 0.03 |

**Point 3:** Although the manuscript is written well and results are well displayed, it would be useful to maintain the colour coding of each reanalysis/observational datasets throughout the figures to avoid confusion

*Response:*

Corrected. In the new version, the ULS/ASPeCt data is colored in black and ICESat-1/Envisat is colored in red. Please see the new Figure 2 and 3. Besides, these color codes are applied to ITDs in the new Figure 5.

[Figure]

New Figure 2: a) Probability density distributions (PDF) of monthly sea-ice thickness from ULS and four reanalyses at the 13 ULS locations of the Weddell Sea. b) Normalized Taylor diagram for monthly sea ice thickness of four reanalyses as well as Envisat and ICESat-1 with respect to the sea-ice thickness from upward-looking sonar from 1990 to 2008 in the Weddell Sea. The green dashed lines indicate the normalized root-mean-square error (RMSE).

[Figure]

New Figure 3: Same as Figure 1b, but for the four sub-regions: a) Antarctic Peninsula, b) central Weddell Sea, c) Southern Coast, and d) eastern Weddell Sea.

[Figure]

New Figure 5: The variation of monthly ice thickness distribution from GECCO2 (blue), SOSE (cyan), NEMO-EnKF (green), GIOMAS (pink) and ICESat-1 (red) in Autumn-FM (left), Winter-MJ (middle) and Spring-ON (right). The colored dots represent the modal ice thickness. In order to make the histogram plots readable, different reanalyses has different x range.

**Point 4:** It is clear that the reanalyses underestimate the sea ice thickness distribution when compared to ICESat-1 but maybe some attention should be given to the description of this comparison in the results. "ICESat-1 thickness is much thicker than that of the reanalyses except GIOMAS in Spring-ON" – is this entirely accurate? GECCO2 also has two instances where the modes are similar when compared to ICESat-1, in Spring-ON (2007) the mode is higher in GECCO2, Spring-ON (2006) it appears to be the same. NEMO-EnkF also has two examples in Autumn-FM where the modes are higher (2006) and similar (2005). Is GIOMAS in Spring-ON really that notable? I am a little confused by the plots in Figure 5 from visual inspection - why does the ICESat-1 thickness distribution change in the same season and year for the SOSEcomparison? i.e. the ICESat-1 distributions seem to stay the same in the PDFs in the same year/season for the other plots but the distribution is shifted in the SOSE plots. Is this to do with some different sampling from different ge ographical coverage of the reanalyses? In addition, why is 2007/2008 omitted for Winter-MJ and 2008 for SpringON? Was this decision driven by a lack of ICESat-1 data for comparison? It is stated in the text (L264) that the 'we compare sea-ice thickness from four renanalyses: : :with that from ICESat-1 for the period from 2005 to 2008' but this does not appear to be the case in the corresponding figure. In Table 2 the ICESat-1 measurement periods are described and 2007 (Winter-MJ) does not have a '-' indicating the data is absent but instead 'Winter-MJ' is written. Also 2004 is shown but does not appear to be part of the described analysis, is there a reason for this? My main point related to this section of the study is that there seem to be some discrepancies in how the data is described and how it is presented in figures. This needs to be looked at and all data and descriptions must be consistent

*Response:* We are very sorry as we made a mistake in the old Table 2. In fact, there are no ICESat-1 measurements in the 2007 winter-MJ. We only compare four reanalyses with ICESat-1 from 2005 to 2008 to fit for SOSE time coverage as well as to be comparable with Figure 6. In order to make the histogram plots in Figure 5 more readable, we use a different x range for histogram plots of different reanalyses. The key point of Figure 5 is the location of modal ice thickness, which is the positions of colored circle dots. In addition, the descriptions of the differences between reanalyses and ICESat-1 were improved in the new version based on the new ranking results and the new monthly averaged approach.

**Point 5:** I would expect that the community would look to evaluations like this to understand what reanalyses could be useful for supporting their own work. As the manuscript currently reads, it is difficult to digest and really understand the limitations of each of the reanalyses. It may provide some clarity and assist the readers understanding of the results to have a table with the key parameters the authors are trying to evaluate (including but not limited to thickness, relationship between mean/mode, min/max thickness accuracy, spatial accuracy, sea ice growth/seasonal evolution of thickness, open ocean vs. coastal regimes, ice motion –divergence/convergence) and a score evaluating how well they have performed. This is not an explicit request, but just a suggestion for the authors to consider in order to improve the communication of important information from this work.

*Response:* Thanks and please see the response in Point 2.

**Minor Comments:**

**1 L28:** 'crucial component of the Earth system', understand what authors mean but perhaps more

specific 'climate system' for example.

*Response:* Corrected.

**2 L89:** add 'a' between 'introduce' and 'sea-ice'.

*Response:* Corrected.

**3 Section 2.1 ~ L110 and L130:** should a spatial resolution be reported for GECCO2/GIOMAS as is given for the other reanalyses? I see they are in Table 1 but why report some resolutions in the text and not others?

*Response:* We added some description on the resolution of GECCO2 and GIOMAS in the new version.

**4 L118 and L124:** 'â ¸Aˇ r' used in one instance and 'degrees' in another, perhaps adopt one standard

*Response:* we now use """ in Line 124.

**5 L136:** to be absolutely accurate perhaps reword '(the part above the sea level)' to '(combined ice and snow height above local sea level)'.

*Response:* Corrected.

**6 L140:** 'suggested by Worby'? Is a complete reference available?

*Response:* Corrected. The complete reference is Kern et al. (2016) in Line 141. In the new version, we deleted "suggested by Worby" in Line 140.

**7 L149:** I understand that the limitations of radar altimeters are not the focus of this study but the complexity of the technique/its limitations in the Antarctic are understated by these few sentences. Perhaps include reference to other studies highlighting this to provide the reader with some context if they require it. This takes me to another point, it doesn't appear CS-2 is used in the analysis, why is it described in the data section?

*Response:* Corrected. On the one hand, we added "the uncertainties of the radar altimeter can result from the inaccuracy snow-ice interface and snow-ice formation (Willatt et al., 2010), and also the surface type mixing and surface roughness (Schwegmann et al., 2016; Paul et al., 2018; Tilling et al., 2019)." On the other hand, we will delete the description about CS2 in the Antarctic since we have not used them.

Ref:

1.Willatt, R. C., Giles, K. A., Laxon, S. W., Stone-Drake, L., and Worby, A. P.: Field Investigations of Ku-Band Radar Penetration into Snow Cover on Antarctic Sea Ice, IEEE Trans. Geosci. Remote Sens., 48, 365–372, https://doi.org/10.1109/TGRS.2009.2028237, 2010.

2.Schwegmann, S., Rinne, E., Ricker, R., Hendricks, S., and Helm, V.: About the consistency between Envisat and CryoSat-2 radar freeboard retrieval over Antarctic sea ice, The Cryosphere, 10, 1415–1425, https://doi.org/10.5194/tc-10-1415-2016, 2016.

3.Paul, S., Hendricks, S., Ricker, R., Kern, S., and Rinne, E.: Empirical parametrization of Envisat freeboard retrieval of Arctic and Antarctic sea ice based on CryoSat-2: progress in the ESA Climate Change Initiative, The Cryosphere, 12, 2437–2460, https://doi.org/10.5194/tc-12-2437-2018, 2018.

4.Tilling, R., Ridout, A., and Shepherd, A.: Assessing the Impact of Lead and Floe Sampling on Arctic Sea Ice Thickness Estimates from Envisat and CryoSat-2, J. Geophys. Res., 124,

7473–7485,

**8 L149:** More accurate to say 'the radar altimeter is expected to' (and then provide relevant references) as opposed to 'the radar altimeter can measure'

*Response:* Corrected.

**9 L159:** Use acronym 'ULS' once it is provided and throughout manuscript use acronyms/abbreviations once they are supplied e.g. L199 'Antarctic Peninsula' to 'AP' as it is shortened on L195

*Response:* Corrected. In the new version, we use "ULS" instead of "upward looking sonar" in Line 82, 97, 158, 159, 192 and 193. We use "AP" instead of "Antarctic Peninsula" in Line 195, 199, 220, 236, 292, 338 and 384. We use "CWS" instead of "central Weddell Sea" in Line 200, 222, 239, 242, 292 and 381. We use "EWS" instead of "eastern Weddell Sea" in Line 199, 239, 241, 243, 258, 296, 317, 340, 345 and 379. We use "SC" instead of "southern coast" in Line 200 and 226.

**10 L164:** I don't think 'skilful' is appropriate here, perhaps 'accurate' or 'approximates thickness well' or something similar

*Response:* Corrected.

**11 L167:** What are these uncertainties? 5 cm/8cm/18 cm etc? Are they a spread around the mean +/- 5 cm or direct positive deviations from other reference measurements? If so are there references for these expected accuracies?

*Response:* Following Behrendt et al. (2013), the accuracy of the ice draft is $\pm 5$ cm in the freezing/melting seasons and $\pm 12$ cm in winter. Then, based on the linear regression function (Eq. 1 in Line 162) between ice draft and ice thickness, the accuracy of ULS ice thickness is $\pm 8$ cm in the freezing/melting seasons and $\pm 18$ cm in winter.

**12 Figure 1**: Standard deviation is abbreviated to SD in the figure but to STD in the text, these should be consistent

*Response:* Corrected.

**13 Figure 3**: caption – Capitalise 'southern coast'.

*Response:* Corrected and the new caption will be change to: "Figure 3: Same as Figure 1b, but for the four sub-regions: a) Antarctic Peninsula, b) central Weddell Sea, c) Southern Coast, and d) eastern Weddell Sea."

**14 Figure 5:** Thickness (m) is not actually labelled on the y-axis. Insert '(red)' after second mention of ICESat-1 in the caption.

*Response:* Corrected.

**15 L290:** 'this means that the reanalyses may not well represent coastal processes' – what do the authors specifically mean here in reference to sea ice? Dynamics and convergence against the coast? Interaction with the coastline? Inaccurate bathymetry or coastal currents? Some of the concluding statements are a little vague. I think the study would benefit from being more specific and shed light on the limitations that need to be addressed.

*Response:* Exactly as what the reviewer pointed, these processes could all contribute to the underestimation, however, it is rather difficult to distinguish. For example, we do not know whether each model smooths the bathymetry, and actually how they manipulate the bathymetry can dramatically change the coastal current and sea ice. The only thing clear now is that it

should be lack of accurate sea ice dynamics rather than thermodynamics. We now refined the sentence to be: " this means that the reanalyses may not well represent the coastal sea ice dynamical processes.".

**16 L290:** Why is SOSE not included in the spatially averaged differences here?

*Response:* Sorry, we forgot to list the bias of SOSE. The new description should be "The spatial averaged differences between models and ICESat-1 are -1.30 m (GECCO2), 1.42 m (SOSE), -0.63 m (NEMO-EnKF) and -0.75 m (GIOMAS), respectively".

**17 L325:** 'primary' to 'prime'.

*Response:* Corrected.

**18 L362:** insert 'satellite' before 'altimeters'

*Response:* Corrected.

**19 L369:** 'still' before 'been'.

*Response:* Corrected.

**20 L388:** 'improve' to 'improving'

*Response:* Corrected.

**21 L388:** 'assimilate' to 'assimilating'.

*Response:* Corrected.

22 Figure 6: What time period is this data comparison for? Are they seasonal averages

for all years?

*Response:* The time period is from 2005 to 2008. They cover the irregular months listed in Table 2.

23 In acknowledgements: ICESat-1 data is provided by NASA and NSIDC not ESA. REFERENCED EXISTING REANALYSES

*Response:* Corrected.

---

## Editor Decision (ED1)

Review of The Cryosphere Discussion submission tc-2020-71 'Evaluation of Sea-Ice Thickness from four reanalyses in the Antarctic Weddell Sea'

GENERAL COMMENTS

The accurate large-scale measurement and reporting of trends in Antarctic sea ice thickness are two of the major challenges for contemporary geophysical science. Once developed, giving these trends context amongst the multiple drivers of Antarctic sea ice volume change, from natural and anthropogenic forcing over multi-decadal timescales will present and even greater challenge. Given their limited temporal resolution, this cannot be achieved using observational datasets alone. This manuscript evaluates

sea-ice thickness from four reanalyses against observational data in the Weddell Sea sector of the Southern Ocean. Reanalyses can play an important role in providing context for change during the observational period, and this work has impressively pulled together a large amount of data, from a variety of sources to provide an evaluation.

It is interesting to see these results and impressive that monthly sea ice thickness distribution is generally represented as expected in this sea ice sector. The simplification of the real world in these reanalyses is obviously a major concern particularly regarding their resolution, this is not a critique of this work, but a description of the current state of the science. The major limitation appears to be that these reanalyses are struggling to capture the thicker end of the sea ice distribution in the Weddell Sea (and from other reading, the entire Southern Ocean) and therefore omitting an important segment of the statistical information on the sea ice thickness distribution. This also highlights the fact that they are failing to, and often provide no attempt to simulate sea ice dynamics, a process at the core of Antarctic sea ice production. Their sparse resolution and missing physics related to smaller scale deformation processes is a serious limitation. There are clearly major developments to be made before these reanalyses datasets can be used to inform the community about sea ice thickness trends in the Southern Ocean. This work successfully highlights this current limitation.

I find the manuscript meets the set criteria for publication in The Cryosphere but I do have some concern related to the 'Originality' and 'Significance' criteria. Although the particular area of study and reanalysis models are on the whole original, when placed in a wider context, other work has already visited this question (Uotila et al. (2019) is an intercomparison while Massonet et al. (2013); Holland et al. (2014) are reanalysis assessments also comparing results to observational data). It is difficult to gauge how valuable continued comparisons are, especially related to the difference/variability of the physical processes that construct sea ice thickness in the models. For instance, have significant advancements been made in the reanalyses evaluated here, since the publication of other studies to warrant new evaluations? I have no specific recommendation for this. This is perhaps beyond the scope of the expected review process though I feel it is important to highlight it.

With the consideration of this concern above left to the discretion of the editor, I support the publication of the manuscript if the specific points outlined below are incorporated into the work. I would like to place specific emphasis on the tightening up of the results section and an attempt to provide more quantitative conclusions.

SPECIFIC COMMENTS

1. Readers may benefit from a concise explanation of the general principles of sea ice thickness estimates from reanalysis products. It will be difficult for non-experts in this specific field to grasp the processes considered and limitations during the construction of a sea ice thickness reanalysis product. I appreciate this is the point of the references but it is often helpful to provide an insight as part of the text to assist the reader (supported by references).

2. I appreciate it is sometimes difficult to fit all the relevant information into the limited word count of an abstract, but I think the reader (and work) would benefit from some sort of quantified reporting in the abstract. Terms like 'well reproduce' are somewhat subjective, is there a way to effectively provide a quantitative measure in the abstract of how well these reanalysis perform compared to one another and the observations? i.e. report the key results in a quantitative manner. This could in some way be related to a 'score' suggested in point 5 below.

3. Although the manuscript is written well and results are well displayed, it would be useful to maintain the colour coding of each reanalysis/observational datasets throughout the figures to avoid confusion.

4. It is clear that the reanalyses underestimate the sea ice thickness distribution when compared to ICESat-1 but maybe some attention should be given to the description of this comparison in the results. "ICESat-1 thickness is much thicker than that of

the reanalyses except GIOMAS in Spring-ON" – is this entirely accurate? GECCO2 also has two instances where the modes are similar when compared to ICESat-1, in Spring-ON (2007) the mode is higher in GECCO2, Spring-ON (2006) it appears to be the same. NEMO-EnkF also has two examples in Autumn-FM where the modes are higher (2006) and similar (2005). Is GIOMAS in Spring-ON really that notable? I am a little confused by the plots in Figure 5 from visual inspection - why does the ICESat-1 thickness distribution change in the same season and year for the SOSE comparison? i.e. the ICESat-1 distributions seem to stay the same in the PDFs in the same year/season for the other plots but the distribution is shifted in the SOSE plots. Is this to do with some different sampling from different geographical coverage of the reanalyses? In addition, why is 2007/2008 omitted for Winter-MJ and 2008 for Spring-ON? Was this decision driven by a lack of ICESat-1 data for comparison? It is stated in the text (L264) that the 'we compare sea-ice thickness from four renanalyses…with that from ICESat-1 for the period from 2005 to 2008' but this does not appear to be the case in the corresponding figure. In Table 2 the ICESat-1 measurement periods are described and 2007 (Winter-MJ) does not have a '-' indicating the data is absent but instead 'Winter-MJ' is written. Also 2004 is shown but does not appear to be part of the described analysis, is there a reason for this? My main point related to this section of the study is that there seem to be some discrepancies in how the data is described and how it is presented in figures. This needs to be looked at and all data and descriptions must be consistent.

5. I would expect that the community would look to evaluations like this to understand what reanalyses could be useful for supporting their own work. As the manuscript currently reads, it is difficult to digest and really understand the limitations of each of the reanalyses. It may provide some clarity and assist the readers understanding of the results to have a table with the key parameters the authors are trying to evaluate (including but not limited to thickness, relationship between mean/mode, min/max thickness accuracy, spatial accuracy, sea ice growth/seasonal evolution of thickness, open ocean vs. coastal regimes, ice motion –divergence/convergence) and a score

evaluating how well they have performed. This is not an explicit request, but just a suggestion for the authors to consider in order to improve the communication of important information from this work.

TECHNICAL COMMENTS

The temporal span of this investigation isn't immediately clear. Please make clear by including the time frame of the inter-comparison in the abstract and introduction (the analysis overlaps are written at the beginning of Section 4). I understand it is intermittent given the different lengths of the observational datasets and different reanalyses but some indication via a well worded summary, early in the manuscript would be useful.

L28 – 'crucial component of the Earth system', understand what authors mean but perhaps more specific 'climate system' for example.

L89 – add 'a' between 'introduce' and 'sea-ice'.

Section 2.1 ∼ L110 and L130 - should a spatial resolution be reported for GECCO2/GIOMAS as is given for the other reanalyses? I see they are in Table 1 but why report some resolutions in the text and not others?

L118 and L124 – 'âĄř' used in one instance and 'degrees' in another, perhaps adopt one standard.

L136 – to be absolutely accurate perhaps reword '(the part above the sea level)' to '(combined ice and snow height above local sea level)'.

L140 – 'suggested by Worby'? Is a complete reference available?

L149 – I understand that the limitations of radar altimeters are not the focus of this study but the complexity of the technique/its limitations in the Antarctic are understated by these few sentences. Perhaps include reference to other studies highlighting this to provide the reader with some context if they require it. This takes me to another point,

it doesn't appear CS-2 is used in the analysis, why is it described in the data section?

L149 – More accurate to say 'the radar altimeter is expected to' (and then provide relevant references) as opposed to 'the radar altimeter can measure'.

L159 – Use acronym 'ULS' once it is provided and throughout manuscript use acronyms/abbreviations once they are supplied e.g. L199 'Antarctic Peninsula' to 'AP' as it is shortened on L195.

L164 – I don't think 'skilful' is appropriate here, perhaps 'accurate' or 'approximates thickness well' or something similar.

L167 – What are these uncertainties? 5 cm/8cm/18 cm etc? Are they a spread around the mean +/- 5 cm or direct positive deviations from other reference measurements? If so are there references for these expected accuracies?

Figure 1 – Standard deviation is abbreviated to SD in the figure but to STD in the text, these should be consistent.

Figure 3 caption – Capitalise 'southern coast'.

Figure 5 – Thickness (m) is not actually labelled on the y-axis. Insert '(red)' after second mention of ICESat-1 in the caption.

L290 – 'this means that the reanalyses may not well represent coastal processes' – what do the authors specifically mean here in reference to sea ice? Dynamics and convergence against the coast? Interaction with the coastline? Inaccurate bathymetry or coastal currents? Some of the concluding statements are a little vague. I think the study would benefit from being more specific and shed light on the limitations that need to be addressed.

L290 – Why is SOSE not included in the spatially averaged differences here?

L325 – 'primary' to 'prime'.

L362 – insert 'satellite' before 'altimeters'.

L369 – 'still' before 'been'.

L388 – 'improve' to 'improving'.

L388 – 'assimilate' to 'assimilating'.

Figure 6 – What time period is this data comparison for? Are they seasonal averages for all years?

In acknowledgements – ICESat-1 data is provided by NASA and NSIDC not ESA.

REFERENCED EXISTING REANALYSES

Holland, P. R., N. Bruneau, C. Enright, M. Losch, N. T. Kurtz, and R. Kwok, 2014: Modeled Trends in Antarctic Sea Ice Thickness. J. Climate, 27, 3784–3801, https://doi.org/10.1175/JCLI-D-13-00301.1.

Massonnet, F., Mathiot, P., Fichefet, T., Goosse, H., König Beatty, C., Vancoppenolle, M., and Lavergne, T.: A model reconstruction of the Antarctic sea ice thickness and volume changes over 1980–2008 using data assimilation, Ocean Modelling, 64, 67-75, 10.1016/j.ocemod.2013.01.003, 2013.

Uotila, P., Goosse, H., Haines, K. et al. An assessment of ten ocean reanalyses in the polar regions. Clim Dyn 52, 1613–1650 (2019). https://doi.org/10.1007/s00382-018-4242-z.
* * *
[Figure]

I would like to start by pointing out that I was asked to serve as a reviewer in June.

This manuscript evaluates the Southern Ocean sea ice thickness produced by four reanalyses against observations from AUVs, ships and satellites. The manuscript is quite good honestly. Sure it is not how I would have written it, so that I regularly took note of "[whatever] is missing" that I erased after reading the information a few lines later, but nothing that really impairs understanding. There's a lot of figures, but they all have a reason to be here.

I have two somewhat methodological points that I would like to see addressed, and a series of comments to improve the readability, but I consider that it should not require

a lot of work. Hence my evaluation "minor revisions".

1) The regions On Fig 1, you present the four regions into which you split the Weddell Sea, and that you analyse in Fig 3. You base that split on data from ULS, but you present only their mean, not the uncertainty attached to it. I am particularly surprised that 210 and 212 would be in different regions. So at least on Fig 1b, add the errors bars. Then modify the region split if needed.

2) The more recent time period and long term perspective Most of the analysis is performed on the time period common to all four reanalyses (late 2000s), which I understand. Unfortunately, it is a bit old and short. Southern Ocean sea ice has behaved very differently since. So please, include a short extra subsection dedicated to comparing GIOMAS and GECCO to SICCI (CryoSat2 at least) and APP. Ideally, also add something about trends in these reanalyses.

Now for the more minor comments, in order of appearance:

Line 109: say that all the information to come is summarised in Table 1. Try to write this entire section in a more structured manner, giving the same information about all four products (at least time period and resolution).

Line 140: you mention ASPeCT now, but only introduce the product line 170.

Fig 1a: add the lines separating the four regions plotted there too Fig 1b: see comment above, add the error bars, and potentially modify your region division accordingly.

Fig 3d: why is the correlation negative for Envisat? What happens? Is the bias mostly in summer or winter?

Fig 4b-d: why are you showing different thickness bands for different products? They are not even the thicknesses you comment on in the text. Please show only one range, so that the reader can compare the reanalyses.

Table 3: have you checked whether the reanalyses are correlated with each other? It

is suspicious that they all seem to have similar biases when compared against ICESat.

Fig 7: present it like Fig 6, as difference against reference rather than actual values. This way, we can compare with Fig 6 (alternatively, present Fig 6 like Fig 7).

Fig 8 (and text corresponding): since the sea ice concentration is about right, and that all reanalyses present similar biases in thickness when compared to satellite retrievals, can it be that the thickness retrievals are the ones that are not perfect yet? Sea ice concentration retrieval is after all more mature.

Line 331: you meant to refer to Fig 8 here.

Line 345/Fig 9: I know you write that you will not investigate the reasons for biases in the reanalyses, but I find the north sea ice of GIOMAS in winter/spring surprising. Is the reanalysis known for having too fast an Antarctic Circumpolar Current? Or is the ice too thin/mobile?

Line 342-346: you forgot to refer to Fig 9 here. The caption of Fig 9 refers to itself instead of Fig 8 by the way.

Table 5: the units need to be fixed. Indicate the net flux in the reference product as well (at least in the caption).

Line 373: not "sea ice ocean models", reanalyses. Sea ice ocean models have their own series of problems, but that's beyond the scope of this review.
* * *
[Figure]

In this study, Shi et al. evaluate sea ice thickness (SIT) in the Weddell Sea from four reanalyses of coupled ocean and sea ice models, against two in-situ observations and two remote sensing datasets. Their results show that these reanalyses have limited success compared with the observations, and they stress the importance of sea ice motion and deformation on the SIT simulations. Modeling and observations of the Antarctic SIT, even for a single region such as the Weddell Sea, are very challenging. This study provides useful insights not only in the SIT reanalyses but also in the SIT observations, which will be a benefit to the sea ice research community. The Introduction, and Data and methods are well written. However, the Results part needs significant improvement. I suggest the manuscript accepted after major revision.

General Comments:

1. There is inconsistency during the comparison in terms of the data. In the "Data and methods" part, the authors state "Comparison are made using monthly means", however, when in 3.3 Comparison with sea-ice thickness from ICESat-1, they are using seasonal mean. This inconsistency must be fixed. It will be much better that the authors describe how they make the comparison in the exact sections.

2. Section 3.1. It remains unclear what kind of mooring data are using here. According to the statement "The aggregate temporal span of ULS observations in AP, CWS, SC and EWS is 148, 79, 185 and 272 months", and consider the numbers of the mooring in these regions, there should be large difference in the mooring data regarding the time duration. I suggest the authors add a plot in their Figure 1 showing the temporal evolution of the mooring observed SIT that are actually used in their comparisons.

3. Figure 4. Not sure why the authors use SITs from different locations in the three reanalyses (Figs. 4b-d). This means they also use different ASPeCT SIT when comparing with the different reanalyses. What can we infer from such different comparisons? I suggest the authors use a consistent comparison: Use the same ASPeCT SIT, with reanalysis SITs interpolated to the same time and same location.

4. Section 3.3. It is not clear what kind of manipulations here for the reanalysis SIT. The authors state "we use October and November to represent spring . . .". However, according to Table 2, the ICESat-1 measurement is irregular, and no full month measurements. Do the authors use the same dates as the observations, or just simply use the full two-month reanalysis data? As the authors here try to compare the mean, it is very important to compare the exact corresponding data in terms of time and locations. Also the authors need give a test with confidence level for the comparison.

5. Line 280-281. "Compared with ICESat-1, only NEMO-EnKF has a similar variation of modal ice thickness from Autumn-FM to Spring-ON, while GECCO2, SOSE and GIOMAS have monotonically increasing model ice thickness". It seems to me 2005 &

2006 for SOSE, and 2006 for GIOMAS have similar seasonal variations in Figure 5. Table 4 looks somewhat misleading as its modal SITs not necessary in the same year.

Specific Comments:

1. Line 40. Not clear. Rephrase it.

2. Line 62. "... it is also reported to have uncertainties due to ...". All observations have uncertainties. Perhaps more important to note what kind of uncertainty it is.

3. Line 104. "all available daily records around specified model grids are averaged monthly". How far away from that model grid?

4. Lines 110 & 114. I think the adjoint method and 4-D Var method are the same here. Can the authors give a brief of their differences?

5. Line 195. "Figure 1b", better remove "b".

6. Line 217. "It is noted that the relatively short ICESat-1 record (13 months) limits the accuracy of this assessment". Not sure how the authors arrive to this statement.

7. Line 233. "Envisat has the lowest CC (-0.19) and highest RMSE (2.06 m) among all data sets, and its STD is comparable with GIOMAS". These numbers are almost unbelievable. The authors have any explanation of this? Is there any reference supporting similar findings? Or is there some error in the manipulation of the data?

8. Line 238. "But in the regions with large amounts of newly formed ice (the central Weddell Sea and the eastern Weddell Sea), SOSE tends to underestimate sea-ice thickness with lower STDs than the other reanalyses". It looks to me no data here to support that SOSE underestimates SIT.

9. Line 331. "Figure 7" should be "Figure 8"?